# DYNAMIC OF STOCHASTIC GRADIENT DESCENT WITH STATE-DEPENDENT NOISE

## ABSTRACT

Stochastic gradient descent (SGD) and its variants are mainstream methods to train deep neural networks. Since neural networks are non-convex, more and more works study the dynamic behavior of SGD and its impact to generalization, especially the escaping efficiency from local minima. However, these works make the over-simplified assumption that the distribution of gradient noise is state-independent, although it is state-dependent. In this work, we propose a novel power-law dynamic with state-dependent diffusion to approximate the dynamic of SGD. Then, we prove that the stationary distribution of power-law dynamic is heavy-tailed, which matches the existing empirical observations. Next, we study the escaping efficiency from local minimum of power-law dynamic and prove that the mean escaping time is in polynomial order of the barrier height of the basin, much faster than exponential order of previous dynamics. It indicates that SGD can escape deep sharp minima efficiently and tends to stop at flat minima that have lower generalization error. Finally, we conduct experiments to compare SGD and power-law dynamic, and the results verify our theoretical findings.

## 1 INTRODUCTION

Deep learning has achieved great success in various AI applications, such as computer vision, natural language processing, and speech recognition (He *et al.*, 2016b; Vaswani *et al.*, 2017; He *et al.*, 2016a). Stochastic gradient descent (SGD) and its variants are the mainstream methods to train deep neural networks, since they can deal with the computational bottleneck of the training over large-scale datasets (Bottou & Bousquet, 2008).

Although SGD can converge to the minimum in convex optimization (Rakhlin *et al.*, 2012), neural networks are highly non-convex. To understand the behavior of SGD on non-convex optimization landscape, on one hand, researchers are investigating the loss surface of the neural networks with variant architectures (Choromanska *et al.*, 2015; Li *et al.*, 2018b; He *et al.*, 2019b; Draxler *et al.*, 2018; Li *et al.*, 2018a); on the other hand, researchers illustrate that the noise in stochastic algorithm may make it escape from local minima (Keskar *et al.*, 2016; He *et al.*, 2019a; Zhu *et al.*, 2019; Wu *et al.*, 2019a; HaoChen *et al.*, 2020). It is clear that whether stochastic algorithms can escape from poor local minima and finally stop at a minimum with low generalization error is crucial to its test performance. In this work, we focus on the dynamic of SGD and its impact to generalization, especially the escaping efficiency from local minima.

To study the dynamic behavior of SGD, most of the works consider SGD as the discretization of a continuous-time dynamic system and investigate its dynamic properties. There are two typical types of models to approximate dynamic of SGD. (Li *et al.*, 2017; Zhou *et al.*, 2019; Liu *et al.*, 2018; Chaudhari & Soatto, 2018; He *et al.*, 2019a; Zhu *et al.*, 2019; Hu *et al.*, 2019; Xie *et al.*, 2020) approximate the dynamic of SGD by Langevin dynamic with constant diffusion coefficient and proved its escaping efficiency from local minima. These works make over-simplified assumption that the covariance matrix of gradient noise is constant, although it is state-dependent in general. The simplified assumption makes the proposed dynamic unable to explain the empirical observation that the distribution of parameters trained by SGD is heavy-tailed (Mahoney & Martin, 2019). To model the heavy-tailed phenomenon, Simsekli *et al.* (2019); Şimşekli *et al.* (2019) point that the variance of stochastic gradient may be infinite, and they propose to approximate SGD by dynamic driven by $\alpha$-stable process with the strong infinite variance condition. However, as shown in the work (Xie

*et al.*, 2020; Mandt *et al.*, 2017), the gradient noise follows Gaussian distribution and the infinite variance condition does not satisfied. Therefore it is still lack of suitable theoretical explanation on the implicit regularization of dynamic of SGD.

In this work, we conduct a formal study on the (state-dependent) noise structure of SGD and its dynamic behavior. First, we show that the covariance of the noise of SGD in the quadratic basin surrounding the local minima is a quadratic function of the state (i.e., the model parameter). Thus, we propose approximating the dynamic of SGD near the local minimum using a stochastic differential equation whose diffusion coefficient is a quadratic function of state. We call the new dynamic *power-law dynamic*. We prove that its stationary distribution is power-law $\kappa$ distribution, where $\kappa$ is the signal to noise ratio of the second order derivatives at local minimum. Compared with Gaussian distribution, power-law $\kappa$ distribution is heavy-tailed with tail-index $\kappa$. It matches the empirical observation that the distribution of parameters becomes heavy-tailed after SGD training without assuming infinite variance of stochastic gradient in (Simsekli *et al.*, 2019).

Second, we analyze the escaping efficiency of power-law dynamic from local minima and its relation to generalization. By using the random perturbation theory for diffused dynamic systems, we analyze the mean escaping time for power-law dynamic. Our results show that: (1) Power-law dynamic can escape from sharp minima faster than flat minima. (2) The mean escaping time for power-law dynamic is only in the polynomial order of the barrier height, much faster than the exponential order for dynamic with constant diffusion coefficient. Furthermore, we provide a PAC-Bayes generalization bound and show power-law dynamic can generalize better than dynamic with constant diffusion coefficient. Therefore, our results indicate that the state-dependent noise helps SGD to escape from sharp minima quickly and implicitly learn well-generalized model.

Finally, we corroborate our theory by experiments. We investigate the distributions of parameters trained by SGD on various types of deep neural networks and show that they are well fitted by power-law $\kappa$ distribution. Then, we compare the escaping efficiency of dynamics with constant diffusion or state-dependent diffusion to that of SGD. Results show that the behavior of power-law dynamic is more consistent with SGD.

Our contributions are summarized as follows: (1) We propose a novel power-law dynamic with state-dependent diffusion to approximate dynamic of SGD based on both theoretical derivation and empirical evidence. The power-law dynamic can explain the heavy-tailed phenomenon of parameters trained by SGD without assuming infinite variance of gradient noise. (2) We analyze the mean escaping time and PAC-Bayes generalization bound for power-law dynamic and results show that power-law dynamic can escape sharp local minima faster and generalize better compared with the dynamics with constant diffusion. Our experimental results can support the theoretical findings.

## 2 BACKGROUND

In empirical risk minimization problem, the objective is $L(w) = \frac{1}{n}\sum_{i=1}^{n}\ell(x_i, w)$, where $x_i, i = 1, \cdots, n$ are $n$ *i.i.d.* training samples, $w \in \mathbb{R}^d$ is the model parameter, and $\ell$ is the loss function. Stochastic gradient descent (SGD) is a popular optimization algorithm to minimize $L(w)$. The update rule is $w_{t+1} = w_t - \eta \cdot \tilde{g}(w_t)$, where $\tilde{g}(w_t) = \frac{1}{b}\sum_{x \in S_b} \nabla_w \ell(x, w_t)$ is the minibatch gradient calculated by a randomly sampled minibatch $S_b$ of size $b$ and $\eta$ is the learning rate. The minibatch gradient $\tilde{g}(w_t)$ is an unbiased estimator of the full gradient $g(w_t) = \nabla L(w_t)$, and the term $(g(w_t) - \tilde{g}(w_t))$ is called *gradient noise* in SGD.

**Langevin Dynamic** In (He *et al.*, 2019a; Zhu *et al.*, 2019), the gradient noise is assumed to be drawn from Gaussian distribution according to central limit theorem (CLT), i.e., $g(w) - \tilde{g}(w) \sim \mathcal{N}(0, C)$, where covariance matrix $C$ is a constant matrix for all $w$. Then SGD can be regarded as the numerical discretization of the following Langevin dynamic,

$$dw_t = -g(w_t)dt + \sqrt{\eta}C^{1/2}dB_t, \tag{1}$$

where $B_t$ is a standard Brownian motion in $\mathbb{R}^d$ and $\sqrt{\eta}C^{1/2}dB_t$ is called the diffusion term.

$\alpha$**-stable Process** Simsekli *et al.* (2019) assume the variance of gradient noise is unbounded. By generalized CLT, the distribution of gradient noise is $\alpha$-stable distribution $\mathcal{S}(\alpha, \sigma)$, where $\sigma$ is the $\alpha$-th moment of gradient noise for given $\alpha$ with $\alpha \in (0, 2]$. Under this assumption, SGD is approximated by the stochastic differential equation (SDE) driven by an $\alpha$-stable process.

## 2.1 RELATED WORK

There are many works that approximate SGD by Langevin dynamic and most of the theoretical results are obtained for Langevin dynamic with constant diffusion coefficient. From the aspect of optimization, the convergence rate of SGD and its optimal hyper-parameters have been studied in (Li *et al.*, 2017; He *et al.*, 2018; Liu *et al.*, 2018; He *et al.*, 2018) via optimal control theory. From the aspect of generalization, Chaudhari & Soatto (2018); Zhang *et al.* (2018); Smith & Le (2017) show that SGD implicitly regularizes the negative entropy of the learned distribution. Recently, the escaping efficiency from local minima of Langevin dynamic has been studied (Zhu *et al.*, 2019; Hu *et al.*, 2019; Xie *et al.*, 2020). He *et al.* (2019a) analyze the PAC-Bayes generalization error of Langevin dynamic to explain the generalization of SGD.

The solution of Langevin dynamic with constant diffusion coefficient is Gaussian process, which does not match the empirical observations that the distribution of parameters trained by SGD is a heavy-tailed (Mahoney & Martin, 2019; Hodgkinson & Mahoney, 2020; Gurbuzbalaban *et al.*, 2020). Simsekli *et al.* (2019); Şimşekli *et al.* (2019) assume the variance of stochastic gradient is infinite and regard SGD as discretization of a stochastic differential equation (SDE) driven by an $\alpha$-stable process. The escaping efficiency for the SDE is also shown in (Simsekli *et al.*, 2019).

However, these theoretical results are derived for dynamics with constant diffusion term, although the gradient noise in SGD is state-dependent. There are some related works analyze state-dependent noise structure in SGD, such as label noise in (HaoChen *et al.*, 2020) and multiplicative noise in (Wu *et al.*, 2019b). These works propose new algorithms motivated by the noise structure, but they do not analyze the escaping behavior of dynamic of SGD and the impact to generalization. Wu *et al.* (2018) analyze the escaping behavior of SGD with considering the fluctuations of the second order derivatives and propose the concept linearly stability. In our work, we propose power-law dynamic to approximate SGD and analyze the stationary distribution and the mean escaping time for it.

## 3 APPROXIMATING SGD BY POWER-LAW DYNAMIC

In this section, we study the (state-dependent) noise structure of SGD (in Section 3.1) and propose power-law dynamic to approximate the dynamic of SGD. We first study 1-dimensional power-law dynamic in Section 3.2 and extend it to high dimensional case in Section 3.3.

### 3.1 NOISE STRUCTURE OF STOCHASTIC GRADIENT DESCENT

For non-convex optimization, we investigate the noise structure of SGD around local minima so that we can analyze the escaping efficiency from it. We first describe the quadratic basin where the local minimum is located. Suppose $w^*$ is a local minimum of the training loss $L(w)$ and $g(w^*) = 0$. We name the $\epsilon$-ball $\mathcal{B}(w^*, \epsilon)$ with center $w^*$ and radius $\epsilon$ as a quadratic basin if the loss function for $w \in \mathcal{B}(w^*, \epsilon)$ is equal to its second-order Taylor expansion as $L(w) = L(w^*) + \frac{1}{2}(w - w^*)^T H(w^*)(w - w^*)$. Here, $H(w^*)$ is the Hessian matrix of loss at $w^*$, which is (semi) positive definite.

Then we start to analyze the gradient noise of SGD. The full gradient of training loss is $g(w) = H(w^*)(w - w^*)$. The stochastic gradient is $\tilde{g}(w) = \tilde{g}(w^*) + \tilde{H}(w^*)(w - w^*)$ by Taylor expansion where $\tilde{g}(\cdot)$ and $\tilde{H}(\cdot)$ are stochastic version of gradient and Hessian calculated by the minibatch. The randomness of gradient noise comes from two parts: $\tilde{g}(w^*)$ and $\tilde{H}(w^*)$, which reflects the fluctuations of the first-order and second-order derivatives of the model at $w^*$ over different minibatches, respectively. The following proposition gives the variance of the gradient noise.

**Proposition 1** *For $w \in \mathcal{B}(w^*, \epsilon) \subset \mathbb{R}$, the variance of gradient noise is $\sigma(g(w) - \tilde{g}(w)) = \sigma(\tilde{g}(w^*)) + 2\rho(\tilde{g}(w^*), \tilde{H}(w^*))(w - w^*) + \sigma(\tilde{H}(w^*))(w - w^*)^2$, where $\sigma(\cdot)$ and $\rho(\cdot, \cdot)$ are the variance and covariance in terms of the minibatch.*

From Proposition 1, we can conclude that: (1) The variance of noise is finite if $\tilde{g}(w^*)$ and $\tilde{H}(w^*)$ have finite variance because $\rho(\tilde{g}(w^*), \tilde{H}(w^*)) \leq \sqrt{\sigma(\tilde{g}(w^*)) \cdot \sigma(\tilde{H}(w^*))}$ according to Cauchy–Schwarz inequality. For fixed $w^*$, a sufficient condition for that $\tilde{g}(w^*)$ and $\tilde{H}(w^*)$ have finite variance is that

the training data $x$ are sampled from bounded domain. This condition is easy to be satisfied because the domain of training data are usually normalized to be bounded before training. In this case, the infinite variance assumption about the stochastic gradient in $\alpha$-stable process is not satisfied. (2) The variance of noise is state-dependent, which contradicts the assumption in Langevin dynamic.

**Notations:** For ease of the presentation, we use $C(w), \sigma_g, \sigma_H, \rho_{g,H}$ to denote $\sigma(g(w) - \tilde{g}(w^*))$, $\sigma(\tilde{g}(w^*)), \sigma(\tilde{H}(w^*)), \rho(\tilde{g}(w^*), \tilde{H}(w^*))$ in the following context, respectively. [1]

### 3.2  POWER-LAW DYNAMIC

According to CLT, the gradient noise follows Gaussian distribution if it has finite variance, i.e.,

$$g(w) - \tilde{g}(w) \to_d \mathcal{N}(0, C(w)) \quad as \quad b \to \infty, \tag{2}$$

where $\to_d$ means "converge in distribution". Using Gaussian distribution to model the gradient noise in SGD, the update rule of SGD can be written as:

$$w_{t+1} = w_t - \eta g(w_t) + \eta \xi_t, \quad \xi_t \sim \mathcal{N}(0, C(w)). \tag{3}$$

Eq.3 can be treated as the discretization of the following SDE, which we call it power-law dynamic:

$$dw_t = -g(w_t)dt + \sqrt{\eta C(w)}dB_t. \tag{4}$$

Power-law dynamic characterizes how the distribution of $w$ changes as time goes on. The distribution density of parameter $w$ at time $t$ (i.e., $p(w,t)$) is determined by the Fokker-Planck equation (Zwanzig's type (Guo & Du, 2014)):

$$\frac{\partial}{\partial t}p(w,t) = \nabla p(w,t)g(w) + \frac{\eta}{2} \cdot \nabla \left( C(w) \cdot \nabla p(w,t) \right). \tag{5}$$

The stationary distribution of power-law dynamic can be obtained if we let the left side of Fokker-Planck equation be zero. The following theorem shows the analytic form of the stationary distribution of power-law dynamic, which is heavy-tailed and the tail of the distribution density decays at polynomial order of $w - w^*$. This is the reason why we call the stochastic differential equation in Eq.4 power-law dynamic.

**Theorem 2** *The stationary distribution density for 1-dimensional power-law dynamic (Eq.4) is*

$$p(w) = \frac{1}{Z}(C(w))^{-\frac{H}{\eta\sigma_H}} \exp\left( \frac{H\left( 4\rho_{g,H} \cdot ArcTan\left( C'(w)/\sqrt{4\sigma_H\sigma_g - 4\rho_{g,H}^2} \right) \right)}{\eta\sigma_H\sqrt{4\sigma_H\sigma_g - 4\rho_{g,H}^2}} \right), \tag{6}$$

*where $C(w) = \sigma_g + 2\rho_{g,H}(w-w^*) + \sigma_H(w-w^*)^2$, $Z$ is the normalization constant and $ArcTan(\cdot)$ is the arctangent function.*

We make discussions on property of $p(w)$. The decreasing rate of $p(w)$ as $w$ goes away from the center $w^*$ is mainly determined by the term $C(w)^{-\frac{H}{\eta\sigma_H}}$ (because the function $ArcTan(\cdot)$ is bounded) which is a polynomial function about $w - w^*$. Compared with Gaussian distribution the probability density which follows exponential decreasing rate, power-law distribution is less concentrated in the quadratic basin $\mathcal{B}(w^*, \epsilon)$ and heavy-tailed. We call $\frac{H}{\eta\sigma_H}$ the *tail-index* of $p(w)$ and denote it as $\kappa$ in the following context.

We can conclude that the state-dependent noise results in heavy-tailed distribution of parameters, which matches the observations in (Mahoney & Martin, 2019). Langevin dynamic with constant diffusion can be regarded as special case of power-law dynamic when $\rho_{H,g} = 0$ and $\sigma_H = 0$. In this case, $p(w)$ degenerates to Gaussian distribution. Compared with $\alpha$-stable process, we do not assume infinite variance on gradient noise and demonstrate another mechanism that results in heavy-tailed distribution of parameters.

We empirically observe the covariance matrix around the local minimum of training loss on deep neural networks. The results are shown in Figure.1. Readers can refer more details in Appendix 7.1. We have the following observations: (1) The traces of covariance matrices for the deep neural

---

[1] In the following context, we assume $\sigma_g$ is positive number.

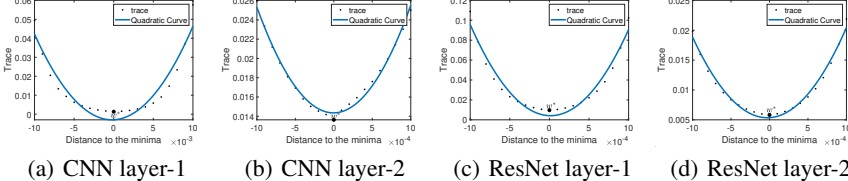

| (a) CNN layer-1 | (b) CNN layer-2 | (c) ResNet layer-1 | (d) ResNet layer-2 |

Figure 1: Trace of covariance matrix of gradient noise in a region around local minimum $w^*$. $w^*$ is selected by running gradient descent with small learning rate till it converges. The number at horizontal axis shows the distance of the point away from $w^*$. **(a),(b):** Results for plain CNN. **(c),(d):**Results for ResNet18.

networks can be well approximated by quadratic curves, which supports Proposition 1. (2) The minimum of the quadratic curve is nearly located at the local minimum $w^*$. It indicates that the coefficient of the first-order term $\rho_{g,H} \approx 0$.

Based on the fact that $\rho_{g,H}$ is not the determinant factor of the tail of the distribution in Eq.6 and the observations in Figure.1, we consider a simplified form of $C(w)$ that $C(w) = \sigma_g + \sigma_H (w - w^*)^2$.

**Corollary 3** *If $C(w) = \sigma_g + \sigma_H (w - w^*)^2$, the stationary distribution of 1-dimensional power-law dynamic (Eq.4) is*

$$p(w) = \frac{1}{Z}(1 + \sigma_H \sigma_g^{-1}(w - w^*)^2)^{-\kappa}, \tag{7}$$

*where $Z$ is the normalization constant and $\kappa = \frac{H}{\eta \sigma_H}$ is the tail-index.*

The distribution density in Eq.7 is known as the power-law $\kappa$ distribution (Zhou & Du, 2014) (It is also named as $q$-Gaussian distribution in (Tsallis & Bukman, 1996)). As $\kappa \to \infty$, the distribution density tends to be Gaussian, i.e., $p(w) \propto \exp(-\frac{H(w-w^*)^2}{\eta \sigma_g})$. Power-law $\kappa$ distribution becomes more heavy-tailed as $\kappa$ becomes smaller. Meanwhile, it produces higher probability to appear values far away from the center $w^*$. Intuitively, smaller $\kappa$ helps the dynamic to escape from local minima faster.

In the approximation of dynamic of SGD, $\kappa$ equals the signal (i.e., $H(w^*)$) to noise (i.e., $\eta \sigma_H$) ratio of second-order derivative at $w^*$ in SGD, and $\kappa$ is linked with three factors: (1) the curvature $H(w^*)$; (2) the fluctuation of the curvature over training data; (3) the hyper-parameters including $\eta$ and minibatch size $b$. Please note that $\sigma_H$ linearly decreases as the batch size $b$ increases.

### 3.3 MULTIVARIATE POWER-LAW DYNAMIC

In this section, we extend the power-law dynamic to $d$-dimensional case. We first illustrate the covariance matrix $C(w)$ of gradient noise in SGD. We use the subscripts to denote the element in a vector or a matrix. We use $\Sigma_g$ to denote the covariance matrix of $\tilde{g}(w^*)$ and assume that $\Sigma_g$ is isotropic (i.e., $\Sigma_g = \sigma_g \cdot I$). We also assume that $Cov(\tilde{H}_i(w^*), \tilde{H}_j(w^*))$ are equal for all $i, j$. It can be shown that $C(w) = \Sigma_g(1 + (w - w^*)^T \Sigma_H \Sigma_g^{-1}(w - w^*))$. Similarly as 1-dimensional case, we omit the first-order term $(w - w^*)$ in $C(w)$. Readers can refer Proposition 10 in Appendix 7.2 for the detailed derivation.

We suppose that the signal to noise ratio of $\tilde{H}(w^*)$ can be characterized by a scalar $\kappa$, i.e., $\eta \Sigma_H = \frac{1}{\kappa} \cdot H(w^*)$. Then $C(w)$ can be written as

$$C(w) = \Sigma_g(1 + \frac{1}{\eta \kappa}(w - w^*)^T H(w^*)\Sigma_g^{-1}(w - w^*)). \tag{8}$$

**Theorem 4** *If $w \in \mathbb{R}^d$ and $C(w)$ has the form in Eq.(8) for $w \in \mathcal{B}(w^*, \epsilon)$. The stationary distribution density of power-law dynamic is*

$$p(w) = \frac{1}{Z}[1 + \frac{1}{\eta \kappa}(w - w^*)^T H(w^*)\Sigma_g^{-1}(w - w^*)]^{-\kappa} \tag{9}$$

*for $w \in \mathcal{B}(w^*, \epsilon)$, where $Z$ is the normalization constant and $\kappa$ satisfies $\eta \Sigma_H = \frac{1}{\kappa} \cdot H(w^*)$.*

**Remark:** The multivariate power-law $\kappa$ distribution (Eq.9) is a natural extension of the 1-dimensional case. Actually, the assumptions on $\Sigma_g$ and $\kappa$ can be replaced by just assuming $\Sigma_g, H(w^*), \Sigma_H$ are codiagonalized. Readers can refer Proposition 11 in Appendix 7.2 for the derivation.

## 4 ESCAPING EFFICIENCY OF POWER-LAW DYNAMIC

In this section, we analyze the escaping efficiency of power-law dynamic from local minima and its relation to generalization. Specifically, we analyze the mean escaping time for $w_t$ to escape from a basin. As shown in Figure.2, we suppose that there are two basins whose bottoms are denoted as $a$ and $c$ respectively and the saddle point $b$ is the barrier between two basins. The barrier height is denoted as $\Delta L = L(b) - L(a)$.

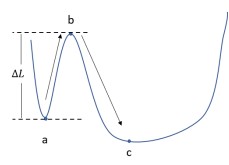

Figure 2

**Definition 5** *Suppose $w_t$ starts at the local minimum $a$, we denote the time for $w_t$ to first reach the saddle point $b$ as $\inf\{t > 0 | w_0 = a, w_t = b\}$. The mean escaping time $\tau$ is defined as $\tau = \mathbb{E}_{w_t}[\inf\{t > 0 | w_0 = a, w_t = b\}]$.*

We first give the mean escaping time for 1-dimensional case in Lemma 6 and then we give the mean escaping time for high-dimensional power-law dynamic in Theorem 7. To analyze the mean escaping time, we take the following assumptions.

**Assumption 1:** The loss function around critical points can be written as $L(w) = L(w^*) + \frac{1}{2}(w - w^*)^T H(w^*)(w - w^*)$, where $w^*$ is a critical point.

**Assumption 2:** The system is in equilibrium near minima, i.e., $\frac{\partial p(w,t)}{\partial t} = 0$.

**Assumption 3:** (Low temperature assumption) The gradient noise is small, i.e., $\eta \sigma_g \ll \Delta L$.

These three assumptions are commonly used in analyzing escaping time (Xie *et al.*, 2020; Zhou & Du, 2014) for a dynamic. Because both $a$ and $b$ are critical points, we can apply Assumption 1 to get the loss surface around them. We put more discussions about the assumptions in Appendix 7.3.2.

We suppose the basin $a$ is quadratic and the variance of noise has the form that $C(w) = \sigma_{g_a} + \sigma_{H_a}(w - a)^2$, which can also be written as $C(w) = \sigma_{g_a} + \frac{2\sigma_{H_a}}{H_a}(L(w) - L(a))$. Furthermore, we suppose that $C(w) = \sigma_{g_a} + \frac{2\sigma_{H_a}}{H_a}(L(w) - L(a))$ on the whole escaping path from $a$ to $b$ (not just near the local minimum $a$). It means that the variance of gradient noise becomes larger as the loss becomes larger. The following lemma gives the mean escaping time of power-law dynamic for 1-dimensional case.

**Lemma 6** *Suppose that Assumption 1-3 are satisfied and $C(w) = \sigma_{g_a} + \frac{2\sigma_{H_a}}{H_a}(L(w) - L(a))$ on the whole escaping path from $a$ to $b$. The mean escaping time of 1-dimensional power-law dynamic is,*

$$\tau = \frac{2\pi}{(1 - \frac{1}{2\kappa})\sqrt{H_a |H_b|}} \left(1 + \frac{2}{\kappa \eta \sigma_{g_a}} \Delta L\right)^{\kappa - \frac{1}{2}}, \tag{10}$$

*where $\kappa = \frac{H_a}{\eta \sigma_{H_a}} > \frac{1}{2}$, $H_a$ and $H_b$ are the second-order derivatives of training loss at local minimum $a$ and at saddle point $b$, respectively.*

The proof of Lemma 6 is based on the results in (Zhou & Du, 2014). We provide a full proof in Appendix 7.3.1. For the dynamic near the saddle point, we just assume that its dynamic is the same as that near the local minimum for simplicity. This assumption is not necessary and we put the extension to more complex dynamic in Appendix 7.3.3.

We summarize the mean escaping time of power-law dynamic and dynamics in previous works in Table 1. Based on the results, we have the following discussions.

**Comparison with other dynamics:** (1) Both power-law dynamic and Langevin dynamic can escape sharp minima faster than flat minima, where the sharpness is measured by $H_a$ and larger $H_a$ corresponds to sharper minimum. Power-law dynamic improves the order of barrier height (i.e., $\Delta L$) from exponential to polynomial compared with Langevin dynamic, which implies a faster escaping efficiency of SGD to escape from deep basin. (2) The mean escaping time for $\alpha$-stable process is independent with the barrier height, but it is in polynomial order of the width of the basin (i.e.,

Table 1: Summary of related works and ours. Here, we only show 1-dimensional result for escaping time in the table for all the three dynamics for ease of the presentation.

| Noise distribution | Dynamic | Stationary solution | Escaping time |
|---|---|---|---|
| $\mathcal{N}(0, \sigma)$ | Langevin | Gaussian | $\mathcal{O}\left(\frac{1}{\sqrt{H_a|H_b|}}\exp\left(\frac{2\Delta L}{\eta\sigma}\right)\right)$ |
| $\mathcal{S}(\alpha, \sigma)$ | $\alpha$-stable | Heavy-tailed | $\mathcal{O}\left(\eta\alpha\cdot\left(\frac{|b-a|}{\eta\sigma}\right)^{\alpha}\right)$ |
| $\mathcal{N}(0, \sigma_g + \sigma_H(w-w^*)^2)$ | Power-law (ours) | Heavy-tailed | $\mathcal{O}\left(\frac{1}{\sqrt{H_a|H_b|}}(1 + \frac{2}{\kappa}\frac{\Delta L}{\eta\sigma_g})^{\kappa-\frac{1}{2}}\right)$ |

width=$|b - a|$). Compared with $\alpha$-stable process, the result for power-law dynamic is superior in the sense that it is also in polynomial order of the width (if $\Delta L \approx O(|b-a|^2)$) and power-law dynamic does not rely on the infinite variance assumption.

Based on Lemma 6, we analyze the mean escaping time for $d$-dimensional case. Under the low temperature condition, the probability density concentrates only along the most possible escaping paths in the high-dimensional landscape. For rigorous definition of most possible escaping paths, readers can refer section 3 in (Xie *et al.*, 2020). For simplicity, we consider the case that there is only one most possible escaping path between basin a and basin c. Specifically, the Hessian at saddle point $b$ has only one negative eigenvalue and the most possible escaping direction is the direction corresponding to the negative eigenvalue of the Hessian at $b$.

**Theorem 7** *Suppose that Assumption 1-3 are satisfied. For $w \in \mathbb{R}^d$, we suppose $C(w) = \Sigma_{g_a} + \frac{2}{\eta\kappa}(L(w) - L(a))$ on the whole escaping path from $a$ to $b$ and there is only one most possible path path between basin $a$ and basin $c$. The mean escaping time for power-law dynamic escaping from basin $a$ to basin $c$ is*

$$\tau = \frac{2\pi\sqrt{-\det(H_b)}}{(1-\frac{d}{2\kappa})\sqrt{\det(H_a)}}\frac{1}{|H_{be}|}\left(1 + \frac{1}{\eta\kappa\sigma_e}\Delta L\right)^{\kappa-\frac{1}{2}}, \tag{11}$$

*where $e$ indicates the most possible escaping direction, $H_{be}$ is the only negative eigenvalue of $H_b$, $\sigma_e$ is the eigenvalue of $\Sigma_{g_a}$ that corresponds to the escaping direction, $\Delta L = L(b) - L(a)$, and $\det(\cdot)$ is the determinant of a matrix.*

**Remark:** In $d$-dimensional case, the flatness is measured by $\det(H_a)$. If $H_a$ has zero eigenvalues, we can replace $H_a$ by $H_a^+$ in above theorem, where $H_a^+$ is obtained by projecting $H_a$ onto the subspace composed by the eigenvectors corresponding to the positive eigenvalues of $H_a$. This is because by Taylor expansion, the loss $L(w)$ only depends on the positive eigenvalues and the corresponding eigenvectors of $H_a$, i.e., $L(w) = L(a) + \frac{1}{2}(w-a)^T H_a(w-a) = L(a) + \frac{1}{2}(\mathbb{P}(w-a))^T \Lambda_{H_a^+}\mathbb{P}(w-a)$, where $\Lambda_{H_a^+}$ is a diagonal matrix composed by non-zero eigenvalues of $H_a$ and the operator $\mathbb{P}(\cdot)$ operates the vector to the subspace corresponding to non-zero eigenvalues of $H_a$. Therefore, the dimension $d$ in Theorem 7 can be regarded as the dimension of subspace that is composed by directions with large eigenvalues. It has been observed that most of the eigenvalues in $H$ is very small (Sagun *et al.*, 2016). Therefore, $d$ will not be a large number and power-law dynamic in multi-dimensional case will inherit the benefit of that in 1-dimensional case compared with Langevin dynamic and $\alpha$-stable process.

The next theorem give an upper bound of the generalization error of the stationary distribution of power-law dynamic, which shows that flatter minimum has smaller generalization error.

**Theorem 8** *Suppose that $w \in \mathbb{R}^d$ and $\kappa > \frac{d}{2}$. For $\delta > 0$, with probability at least $1 - \delta$, the stationary distribution of power-law dynamic has the following generalization error bound,*

$$\mathbb{E}_{w\sim p(w), x\sim\mathcal{P}(x)}\ell(w, x) \le \mathbb{E}_{w\sim p(w)}L(w) + \sqrt{\frac{KL(p||p') + \log\frac{1}{\delta} + \log n + 2}{n-1}},$$

*where $KL(p||p') \le \frac{1}{2}\log\frac{\det(H)}{\det(\Sigma_g)} + \frac{Tr(\eta\Sigma_g H^{-1})-2d}{4\left(1-\frac{1}{\kappa}\left(\frac{d}{2}-1\right)\right)} + \frac{d}{2}\log\frac{2}{\eta}$, $p(w)$ is the stationary distribution of $d$-dimensional power-law dynamic, $p'(w)$ is a prior distribution which is selected to be standard*

*Gaussian distribution, and $\mathcal{P}(x)$ is the underlying distribution of data $x$, $\det(\cdot)$ and $Tr(\cdot)$ are the determinant and trace of a matrix, respectively.*

We make the following discussions on results in Theorem 8. For 1-dimensional case, we have if $H > \frac{\eta}{2(1+\frac{1}{2\kappa})}$, KL divergence is decreasing as $H$ decreases. For $d > 1$ and fixed $Tr(\Sigma_g H^{-1})$ and $\det(\Sigma_g)$, the generalization error (i.e., $\mathbb{E}_{w\sim p(w), x\sim\mathcal{P}(x)}\ell(w,x) - \mathbb{E}_{w\sim p(w)}L(w)$) is decreasing as $\det(H)$ decreases, which indicates that flatter minimum has smaller generalization error. Moreover, if $2d > Tr(\eta\Sigma_g H^{-1})$, the generalization error is decreasing as $\kappa$ increases. When $\kappa \to \infty$, the generalization error tends to that for Langevin dynamic. Combining the mean escaping time and the generalization error bound, we can conclude that state-dependent noise makes SGD escape from sharp minima faster and implicitly tend to learn a flatter model which generalizes better.

## 5 EXPERIMENTS

In this section, we conduct experiments to verify the theoretical results. We first study the fitness between parameter distribution trained by SGD and power-law $\kappa$ distribution. Then we compare the escaping behavior for power-law dynamic, Langevin dynamic and SGD.

### 5.1 FITTING PARAMETER DISTRIBUTION USING POWER-LAW DISTRIBUTION

We investigate the distribution of parameters trained by SGD on deep neural networks and use power-law $\kappa$ distribution to fit the parameter distribution. We first use SGD to train various types of deep neural networks till it converge. For each network, we run SGD with different minibatch sizes over the range $\{64, 256, 1024\}$. For the settings of other hyper-parameters, readers can refer Appendix 7.5.2. We plot the distribution of model parameters at the same layer using histogram. Next, we use power-law $\kappa$ distribution to fit the distribution of the parameters and estimate the value of $\kappa$ via the embedded function "$TsallisQGaussianDistribution[]$" in Mathematica software.

We show results for LeNet-5 with MNIST dataset and ResNet-18 with CIFAR10 dataset (LeCun *et al.*, 2015; He *et al.*, 2016b) in this section, and put results for other network architectures in Appendix 7.5.2. In Figure 3, we report the generalization error (i.e., Test error - Training error) and the values of $\kappa$ that best fit the histogram. [2] We have the following observations: (1) The distribution of the parameter trained by SGD can be well fitted by power-law $\kappa$ distribution (blue curve). (2) As the minibatch size becomes larger, $\kappa$ becomes larger. It is because the noise $\sigma_H$ linearly decreases as minibatch size becomes larger and $\kappa = \frac{H}{\eta\sigma_H}$. (3) As $\kappa$ becomes smaller, the generalization error becomes lower. It indicates that $\kappa$ also plays a role as indicator of generalization. These results are consistent with the theory in Section 4.

### 5.2 COMPARISON ON ESCAPING EFFICIENCY

We use a 2-dimensional model to simulate the escaping efficiency from minima for power-law dynamic, Langevin dynamic and SGD. We design a non-convex 2-dimensional function written as $L(w) = \frac{1}{n}\sum_{i=1}^{n}\ell(w-x_i)$, where $\ell(w) = 15\sum_{j=1}^{2}|w_j - 1|^{2.5} \cdot |w_j + 1|^3$ and training data $x_i \sim \mathcal{N}(0, 0.01I_2)$. We regard the following optimization iterates as the numerical discretization of the power-law dynamic, $w_{t+1} = w_t - \eta g(w_t) + \eta\lambda_2\sqrt{1 + \lambda_1(w_t - w^*)^2} \odot \xi$, where $\xi \sim \mathcal{N}(0, I_2)$, $\lambda_1, \lambda_2$ are two hyper-parameters and $\odot$ stands for Hadamard product. Note that if we set $\lambda_1 = 0$, it can be regarded as discretization of Langevin dynamic. We set learning rate $\eta = 0.025$, and we take 500 iterations in each training. In order to match the trace of covariance matrix of stochastic gradient at minimum point $w^*$ with the methods above, $\lambda_2$ is chosen to satisfy $Tr(Cov(\lambda_2\xi)) = Tr(Cov(g(w^*)))$.

We compare the success rate of escaping for power-law dynamic, Langevin dynamic and SGD by repeating the experiments 100 times. To analyze the noise term $\lambda_1$, we choose different $\lambda_1$ and evaluate corresponding success rate of escaping, as shown in Figure.4(c). The results show that: (1) there is a positive correlation between $\lambda_1$ and the success rate of escaping; (2) power-law dynamic can mimic the escaping efficiency of SGD, while Langevin dynamic can not. We then scale the loss

---

[2] The training errors under the six settings are almost zero.

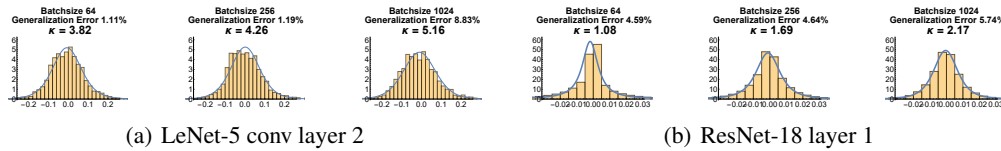

(a) LeNet-5 conv layer 2      (b) ResNet-18 layer 1

Figure 3: Approximating distribution of parameters (trained by SGD) by power-law dynamic. Training batchsize, generalization error (i.e., Test error - Training error) and approximated tail-index $\kappa$ are shown in the title of each plot. **(a):** Results for LeNet-5. **(b):**Results for ResNet-18.

function by $0.9$ to make the minima flatter and repeat all the algorithms under the same setting. The success rate for the scaled loss function is shown in Figure.4(d). We can observe that all dynamics escape flatter minima slower.

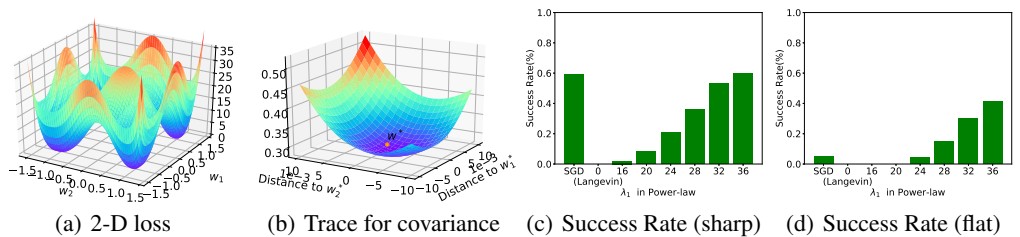

(a) 2-D loss    (b) Trace for covariance    (c) Success Rate (sharp)    (d) Success Rate (flat)

Figure 4: **(a):**Loss surface of $L(w)$ for 2-D model. **(b):**Trace of covariance matrix around minimum $(1, 1)$. **(c)/(d):** Success rate of escaping from the basin of $L(w)$ / $0.9L(w)$ in repeated 100 runs.

## 6   CONCLUSION

In this work, we study the dynamic of SGD via investigating state-dependent variance of the stochastic gradient. We propose power-law dynamic with state-dependent diffusion to approximate the dynamic of SGD. We analyze the escaping efficiency from local minima and the PAC-Bayes generalization error bound for power-law dynamic. Results indicate that state-dependent noise helps SGD escape from poor local minima faster and generalize better. We present direct empirical evidence to support our theoretical findings.This work may motivate many interesting research topics, for example, non-Gaussian state-dependent noise, new types of state-dependent regularization tricks in deep learning algorithms and more accurate characterization about the loss surface of deep neural networks. We will investigate these topics in future work.

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

## 7 APPENDIX

### 7.1 POWER-LAW DYNAMIC AND STATIONARY DISTRIBUTION

**Theorem 9** *(Theorem 2 in main paper) The stationary distribution density for 1-dimensional power-law dynamic (Eq.4) is*

$$p(w) = \frac{1}{Z}(C(w))^{-\frac{H}{\eta \sigma_H}} \exp\left( \frac{H\left(4\rho_{g,H} \cdot ArcTan\left(C'(w)/\sqrt{4\sigma_H \sigma_g - 4\rho_{g,H}^2}\right)\right)}{\eta \sigma_H \sqrt{4\sigma_H \sigma_g - 4\rho_{g,H}^2}} \right),$$

*where $C(w) = \sigma_g + 2\rho_{g,H}(w-w^*) + \sigma_H(w-w^*)^2$, $Z$ is the normalization constant and $ArcTan(\cdot)$ is the arctangent function.*

*Proof:* We denote the function $\frac{H\left(4\rho_{g,H} \cdot ArcTan\left(C'(w)/\sqrt{4\sigma_H \sigma_g - 4\rho_{g,H}}\right)\right)}{\eta \sigma_H \sqrt{4\sigma_H \sigma_g - 4\rho_{g,H}^2}}$ as $h(w)$. According to the Fokker-Planck equation, $p(w)$ satisfies

$$0 = \nabla p(w)g(w) + \frac{\eta}{2} \cdot \nabla \cdot (C(w)\nabla p(w))$$

$$= \nabla \cdot \left[ (p(w) \cdot \nabla L(w)) + \frac{\eta}{2}C(w)\nabla p(w) \right]$$

$$= \nabla \cdot \left[ \frac{\eta}{2}C(w)^{-\frac{H}{\eta \sigma_H}+1} e^{h(w)} \nabla (C(w)^{\frac{H}{\eta \sigma_H}} \cdot e^{-h(w)} \cdot p(w)) \right]$$

Readers can check the third equality by calculating $\nabla(C(w)^{\frac{H}{\eta \sigma_H}} \cdot e^{-h(w)} \cdot p(w))$ with $C(w) = \sigma_g + 2\rho_{g,H}(w-w^*) + \sigma_H(w-w^*)^2$. Because the left side equals zero, we have $C(w)^{\frac{H}{\eta \sigma_H}} \cdot e^{-h(w)} \cdot p(w)$ equals to constant. So $p(w) \propto C(w)^{-\frac{H}{\eta \sigma_H}} \cdot e^{h(w)} \cdot p(w)$. So we can get the conclusion in the theorem. $\square$

**Theorem 10** *(Corollary 3 in main paper) If $C(w) = \sigma_g + \sigma_H(w-w^*)^2$, the stationary distribution density of power-law dynamic is*

$$p(w) = \frac{1}{Z}(1 + \sigma_H \sigma_g^{-1}(w-w^*)^2)^{-\kappa}, \tag{12}$$

*where $Z = \int_w (1 + \sigma_H \sigma_g^{-1}(w-w^*)^2)^{-\kappa} dw$ is the normalization constant and $\kappa = \frac{H}{\eta \sigma_H}$ is the tail-index.*

*Proof:* According to the Fokker-Planck equation, $p(w)$ satisfies

$$0 = \nabla p(w)g(w) + \frac{\eta}{2} \cdot \nabla \cdot (C(w)\nabla p(w))$$

$$= \nabla(p(w) \cdot \nabla L(w)) + \frac{\eta}{2}\nabla \cdot (\sigma_g + \frac{2\sigma_H}{H}(L(w) - L(w^*)))\nabla p(w)$$

$$= \nabla \cdot \frac{\eta}{2}C(w)(1 + \frac{2\sigma_H}{H\sigma_g}(L(w) - L(w^*)))^{\frac{-H}{\eta \sigma_H}} \nabla(1 + \frac{2\sigma_H}{H\sigma_g}(L(w) - L(w^*)))^{\frac{H}{\eta \sigma_H}} p(w)$$

Because the left side equals zero, we have $(1 + \frac{2\sigma_H}{H\sigma_g}(L(w) - L(w^*)))^{\frac{H}{\eta \sigma_H}} p(w)$ equals to constant. So $p(w) \propto (1 + \frac{2\sigma_H}{H\sigma_g}(L(w) - L(w^*)))^{\frac{-H}{\eta \sigma_H}}$. So we can get the conclusion in the theorem. $\square$

We plot the un-normalized distribution density for 1-dimensional power-law dynamics with different $\kappa$ in Figure 5. For the four curves, we set $\beta = 10$. We set $\kappa = 1, 0.5, 0.1, 0$ and use green, red,

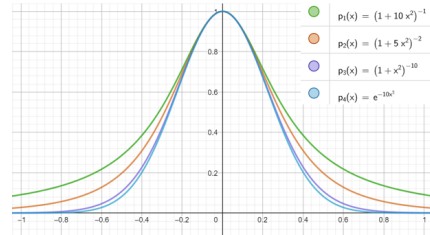

Figure 5: Probability density for power-law dynamic.

purple and blue line to illustrate their corresponding density function, respectively. When $\kappa = 0$, it is Gaussian distribution. From the figure, we can see that the tail for power-law $\kappa$-distribution is heavier than Gaussian distribution.

Actually, for any given time $t$, the distribution $p(w, t)$ for $w_t$ that satisfies power-law dynamic has analytic form, i.e., $p(w, t) \propto (1 + \frac{H}{\eta\kappa\sigma(t)}(w - w(t))^2)^{-\kappa}$, where $w(t) = w^* + (w_0 - w^*)e^{-Ht}$ and $\sigma(t)$ is a function of $\sigma_g$ and $t$. Readers can refer Eq.18 - Eq.23 in (Tsallis & Bukman, 1995) for the detailed expression.

## 7.2 SGD AND MULTIVARIATE POWER-LAW DYNAMIC

The following proposition shows the covariance of stochastic gradient in SGD in $d$-dimensional case. We use the subscripts to denote the elements in a vector or a matrix.

**Proposition 11** *For $w \in \mathbb{R}^d$, we use $C(w)$ to denote the covariance matrix of stochastic gradient $\tilde{g}(w) = \tilde{g}(w^*) + \widetilde{H}(w - w^*)$ and $\Sigma$ to denote the covariance matrix of $\tilde{g}(w^*)$. If $Cov(\tilde{g}_i(w^*), \widetilde{H}_{jk}) = 0, \forall i, j, k$, we have*

$$C_{ij}(w) = \Sigma_{ij} + (w - w^*)^T A^{(ij)}(w - w^*),  \tag{13}$$

*where $\Sigma_{ij} = Cov(\tilde{g}_i(w^*), \tilde{g}_j(w^*))$, $A^{(ij)}$ is a $d \times d$ matrix with elements $A_{ab}^{(ij)} = Cov(\tilde{H}_{ia}, \tilde{H}_{jb})$ with $a \in [d], b \in [d]$.*

Eq.13 can be obtained by directly calculating the covariance of $\tilde{g}_i(w)$ and $\tilde{g}_j(w)$ where $\tilde{g}_i(w) = \tilde{g}_i(w^*) + \sum_{a=1}^d \tilde{H}_{ia}(w_a - w_a^*)$, $\tilde{g}_j(w) = \tilde{g}_j(w^*) + \sum_{b=1}^d \tilde{H}_{jb}(w_b - w_b^*)$.

In order to get a analytic tractable form of $C(w)$, we make the following assumptions: (1) If $\Sigma_{ij} = 0$, $A^{(ij)}$ is a zero matrix; (2) For $\Sigma_{ij} \neq 0$, $\frac{A^{(ij)}}{\Sigma_{ij}}$ are equal for all $i \in [d], j \in [d]$. The first assumption is reasonable because both $\Sigma_{ij}$ and $A^{(ij)}$ reflect the dependence of the derivatives along the $i$-th direction and $j$-th direction. Let $\Sigma_H = \frac{A^{(ij)}}{\Sigma_{ij}}$, $C(w)$ can be written as $C(w) = \Sigma_g(1 + (w - w^*)^T \Sigma_H(w - w^*))$. The $d$-dimensional power-law dynamic is written as

$$dw_t = -H(w - w^*)dt + \sqrt{\eta C(w)}dB_t,  \tag{14}$$

where $C(w) = \Sigma_g(1 + (w - w^*)^T \Sigma_H(w - w^*))$ which is a symmetric positive definite matrix that $C(w)^{1/2}$ exists. The following proposition shows the stationary distribution of the $d$-dimensional power-law dynamic.

**Proposition 12** *Suppose $\Sigma_g, \Sigma_H, H$ are codiagonalizable, i.e., there exist orthogonal matrix $Q$ and diagonal matrices $\Lambda, \Gamma, \Pi$ to satisfy $\Sigma_g = Q^T \Lambda Q, \Sigma_H = Q^T \Gamma Q, H = Q^T \Pi Q$. Then, the stationary distribution of power-law dynamic is*

$$p(w) = \frac{1}{Z}(1 + (w - w^*)^T \Sigma_H(w - w^*))^{-\kappa},  \tag{15}$$

*where $Z$ is the normalization constant and $\kappa = \frac{Tr(H)}{\eta Tr(\Sigma_H \Sigma_g)}$.*

*Proof:* Under the codiagonalization assumption on $\Sigma_g, \Sigma_H, H$, Eq.15 can be rewritten as $dv_t = -\Pi v_t dt + \sqrt{\eta\Lambda(1 + v_t^T\Gamma v_t)}dB_t$ if we let $v_t = Q(w_t - w^*)$.

We use $\phi(v) = \frac{\eta C(v)}{2} = \frac{\eta}{2}\Lambda(1 + v^T\Gamma v)$, the stationary probability density $p(v)$ satisfies the Smoluchowski equation:

$$0 = \sum_{i=1}^{d} \frac{\partial}{\partial v_i}\left(\Pi_i v_i \cdot p(v)\right) + \sum_{i=1}^{d} \frac{\partial}{\partial v_i} \cdot \left(\phi_i(w)\frac{\partial}{\partial v_i}p(v)\right) \tag{16}$$

$$= \sum_{i=1}^{d} \frac{\partial}{\partial v_i}\left(\Pi_i \cdot v_i \cdot p(v)\right) + \sum_{i=1}^{d} \frac{\partial}{\partial v_i} \cdot \left(\frac{\eta\Lambda_i}{2}(1 + v^T\Gamma v)\frac{\partial}{\partial v_i}p(v)\right). \tag{17}$$

According to the result for 1-dimensional case, we have the expression of $p(v)$ is $p(v) \propto (1 + v^T\Gamma v)^{-\kappa}$. To determine the value of $\kappa$, we put $p(v)$ in the Smoluchowski equation to obtain

$$\sum_{i=1}^{d} \Pi_i p(v) - 2\kappa \sum_{i=1}^{d} \Pi_i v_i \cdot \Gamma_i v_i \cdot (1 + v^T\Gamma v)^{-\kappa-1}$$

$$= \sum_{i=1}^{d} \frac{\partial}{\partial v_i}\left(\eta\Lambda_i\kappa(1 + v^T\Gamma v)^{-\kappa} \cdot \Gamma_i v_i\right)$$

$$= \sum_{i=1}^{d} \left(\eta\Lambda_i\kappa(1 + v^T\Gamma v)^{-\kappa} \cdot \Gamma_i\right) - 2\sum_{i=1}^{d} \left(\eta\Lambda_i\kappa^2(1 + v^T\Gamma v)^{-\kappa-1} \cdot (\Gamma_i v_i)^2\right).$$

The we have $\sum_{i=1}^{d} \Pi_i = \eta\kappa\sum_{i=1}^{d} \Lambda_i\Gamma_i$. So we have $\kappa = \frac{Tr(H)}{\eta Tr(\Sigma_H\Sigma_g)}$. $\square$

According to Proposition 11, we can also consider another assumption on $\Sigma_g, \Sigma_H, H$ without assuming their codiagonalization. Instead, we assume (1) If $\Sigma_{ij} = 0$, $A^{(ij)}$ is a zero matrix; (2) For $\Sigma_{ij} \neq 0$, $A^{(ij)}$ are equal for all $i \in [d], j \in [d]$ and we denote $A^{(ij)} = \Sigma_H$. We suppose $\eta \cdot \Sigma_H = \kappa H$. (3) $\Sigma_g = \sigma_g \cdot I_d$ which is isotropic. Under these assumptions, we can get the following theorem.

**Theorem 13** *(Theorem 4 in main paper) If $w$ is $d$-dimensional and $C(w)$ has the form in Eq.(8). The stationary distribution density of multivariate power-law dynamic is*

$$p(w) = \frac{1}{Z}[1 + \frac{1}{\eta\kappa}(w - w^*)^T H\Sigma_g^{-1}(w - w^*)]^{-\kappa} \tag{18}$$

*where $Z = \int_{-\infty}^{\infty}[1 + \frac{1}{\eta\kappa}(w - w^*)^T H\Sigma_g^{-1}(w - w^*)]^{-\kappa}dw$ is the normalization constant.*

The proof for Theorem 12 is similar to that for Proposition 11. Readers can check that $p(w)$ satisfies the Smoluchowski equation.

**An example to illustrate why $C(w)$ is diagonally dominant.** In Theorem 13, $C(w)$ is assumed to be diagonally dominant. Diagonally dominant indicates that the variance of each dimension of $\tilde{g}(w)$ is significantly larger than the covariance of two different dimensions of $\tilde{g}(w)$. Consider a two layer fully-connected linear neural network $f_{w,v}(x) = wvx$ where $w \in \mathbb{R}^{1 \times m}, v \in \mathbb{R}^{m \times d}, x \in \mathbb{R}^d$ and $h(\cdot)$ is the ReLU activation. We consider the regression loss $\ell(w, v) = \frac{1}{2}(y - f_{w,v}(x))^2$. The gradient of $w_i$ and $v_{jk}$ can be written as

$$\frac{\partial\ell(w, v)}{\partial w_i} = (f_{w,v}(x) - y) \cdot v_i x \tag{19}$$

$$\frac{\partial\ell(w, v)}{\partial v_{jk}} = (f_{w,v}(x) - y) \cdot w_j x_k, \tag{20}$$

where $v_i$ denotes the i-th row of matrix $v$. Suppose that the initialization of $w$ and $v$ is: $w_i \overset{i.i.d}{\sim} N(0, \delta_1)$ and $v_{ij} \overset{i.i.d}{\sim} N(0, \delta_2)$. We also assume that $\mathbb{E}x_i = \mathbb{E}x_j = 0$ and $x_i, x_j$ are independent with each other for $i \neq j$ where $x_i$ is the $i$-th dimension. We have

$$\mathbb{E}_{w,v}\frac{\partial\ell(w, v)}{\partial w_i}\frac{\partial\ell(w, v)}{\partial w_j} = \mathbb{E}_{w,v}(f_{w,v}(x) - y)^2 \cdot v_i x \cdot v_j x \tag{21}$$

$$= \mathbb{E}_{w,v}y^2 \cdot v_i x \cdot v_j x + \mathbb{E}_{w,v}\sum_{i=1}^{m}(w_i v_i x)^2 \cdot v_i x \cdot v_j x - 2\mathbb{E}_{w,v}(\sum_{i=1}^{m} yw_i v_i x) \cdot v_i x \cdot v_j x \tag{22}$$

Because the independence of $v_i$, $v_j$ and their expectations are zero, we can obtain $\mathbb{E}_{w,v}\frac{\partial \ell(w,v)}{\partial w_i}\frac{\partial \ell(w,v)}{\partial w_j} = 0$ for $i \neq j$. Similarly, we can get $\mathbb{E}_{w,v}\frac{\partial \ell(w,v)}{\partial w_i}\frac{\partial \ell(w,v)}{\partial v_{jk}} = 0$ and $\mathbb{E}_{w,v}\frac{\partial \ell(w,v)}{\partial v_{j'k'}}\frac{\partial \ell(w,v)}{\partial v_{jk}} = 0$ for $(j,k) \neq (j',k')$.

The above analyses show that the gradients for different dimensions are independent at initialization. It has been observed that many weights are kept random during training because of the over-parameterization Balduzzi *et al.* (2017). So, diagonalization dominant property of $C(w)$ is reasonable.

### 7.3 SUPPLEMENTARY MATERIALS FOR RESULTS IN SECTION 4

#### 7.3.1 PROOF FOR MEAN ESCAPING TIME

**Lemma 14** *(Lemma 6 in main paper) We suppose $C(w) = \sigma_{g_a} + \frac{2\sigma_{H_a}}{H_a}(L(w) - L(a))$ on the whole escaping path from $a$ to $b$. The mean escaping time of the 1-dimensional power-law dynamic is,*

$$\tau = \frac{2\pi}{(1 - \frac{1}{2\kappa})\sqrt{H_a|H_b|}}\left(1 + \frac{2}{\kappa\eta\sigma_{g_a}}\Delta L\right)^{\kappa - \frac{1}{2}}, \tag{23}$$

*where $\kappa = \frac{H_a}{\eta\sigma_{H_a}}$, $H_a$, $H_b$ are the second-order derivatives of training loss at local minimum $a$ and saddle point $b$.*

*Proof:* According to (Van Kampen, 1992), the mean escaping time $\tau$ is expressed as $\tau = \frac{P(w \in V_a)}{\int_\Omega J d\Omega}$, where $V_a$ is the volume of basin $a$, $J$ is the probability current that satisfies

$$-\nabla J(w,t) = \frac{\partial}{\partial w}\left(g(w) \cdot p(w,t)\right) + \frac{\partial}{\partial w}\left(\phi(w)\frac{\partial p(w,t)}{\partial w}\right)$$

$$= \frac{\partial}{\partial w}\left(\phi(w) \cdot \left(1 + \frac{\mu}{\sigma_g}\Delta L(w)\right)^{-\kappa}\frac{\partial\left(\left(1 + \frac{\mu}{\sigma_g}\Delta L(w)\right)^{\kappa}p(w,t)\right)}{\partial w}\right),$$

where $\phi(w) = \frac{\eta}{2}C(w)$ and $\mu = \frac{2\sigma_{H_a}}{H_a}$, $\sigma_g = \sigma_{g_a}$ and $\Delta L(w) = L(w) - L(a)$. Integrating both sides, we obtain $J(w) = -\phi(w) \cdot \left(1 + \frac{\mu}{\sigma_g}\Delta L(w)\right)^{-\kappa}\frac{\partial\left(\left(1 + \frac{\mu}{\sigma_g}\Delta L(w)\right)^{\kappa}p(w,t)\right)}{\partial w}$. Because there is no field source on the escape path, $J(w)$ is fixed constant on the escape path. Multiplying $\phi(w)^{-1} \cdot \left(1 + \frac{\mu}{\sigma_g}\Delta L(w)\right)^{\kappa}$ on both sizes, we have

$$J \cdot \int_a^c \phi(w)^{-1} \cdot \left(1 + \frac{\mu}{\sigma_g}\Delta L(w)\right)^{\kappa}dw = -\int_a^c \frac{\partial\left(\left(1 + \frac{\mu}{\sigma_g}\Delta L(w)\right)^{\kappa}p(w,t)\right)}{\partial w}dw$$

$$= -0 + p(a).$$

Then we get $J = \frac{p(a)}{\int_a^c \phi(w)^{-1} \cdot \left(1 + \frac{\mu}{\sigma_g} \Delta L(w)\right)^\kappa dw}$. As for the term $\int_a^c \phi(w)^{-1} \cdot \left(1 + \frac{\mu}{\sigma_g} \Delta L(w)\right)^{\frac{1}{\kappa}} dw$, we have

$$
\int_a^c \phi(w)^{-1} \cdot \left(1 + \frac{\mu}{\sigma_g} \Delta L(w)\right)^\kappa dw \tag{24}
$$

$$
= \frac{2}{\eta \sigma_g} \int_a^c \left(1 + \frac{\mu}{\sigma_g} \Delta L(w)\right)^{-1+\kappa} dw
$$

$$
= \frac{2}{\eta \sigma_g} \int_c^b \left(1 + \frac{\mu}{\sigma_g}(\Delta L(b) - \frac{1}{2}|H_b|(w-b)^2)\right)^{-1+\kappa} dw
$$

$$
= \frac{2}{\eta \sigma_g} \int_c^b \left(1 + \frac{\mu}{\sigma_g}(\Delta L(b) - \frac{1}{2}|H_b|(w-b)^2)\right)^{-1+\kappa} dw
$$

$$
= \frac{2}{\eta \sigma_g}(1 + \frac{\mu}{\sigma_g}\Delta L(b))^{-1+\kappa} \int_c^b \left(1 - \frac{\mu}{\sigma_g} \cdot \frac{\frac{1}{2}|H_b|(w-b)^2}{1 + \frac{\mu}{\sigma_g}\Delta L(b)}\right)^{-1+\kappa} dw
$$

$$
= \frac{2}{\eta \sigma_g}(1 + \frac{\mu}{\sigma_g}\Delta L(b))^{-1+\kappa} \cdot \left(\frac{\frac{1}{2}\frac{\mu}{\sigma_g}|H_b|}{1 + \frac{\mu}{\sigma_g}\Delta L(b)}\right)^{-1/2} \int_0^1 y^{-1/2}(1-y)^{-1+\kappa} dy
$$

$$
= \frac{2}{\eta \sigma_g}(1 + \frac{\mu}{\sigma_g}\Delta L(b))^{-\frac{1}{2}+\kappa} \sqrt{\frac{2\sigma_g}{\mu|H_b|}} B(\frac{1}{2}, \kappa),
$$

where the third formula is based on the second order Taylor expansion. Under the low temperature assumption, we can use the second-order Taylor expansion around the saddle point $b$.

As for the term $P(w \in V_a)$, we have $P(w \in V_a) = \int_{V_a} p(w) dV = \int_{w \in V_a} p(a)(1 + \frac{\mu}{\sigma_g}\Delta L(w))^{-\kappa} = p(a)\sqrt{\frac{2\sigma_g}{\mu H_a}} B(\frac{1}{2}, \kappa - \frac{1}{2})$, where we use Taylor expansion of $L(w)$ near local minimum $a$. Then we have $\tau = \frac{P(w \in V_a)}{\int_\Omega J d\Omega} = \frac{P(w \in V_a)}{J}$ because $J$ is a constant. Combining all the results, we can get the result in the lemma.

□

**Theorem 15** *(Theorem 7 in main paper) Suppose $w \in \mathbb{R}^d$ and there is only one most possible path path between basin $a$ and the outside of basin $a$. The mean escaping time for power-law dynamic escaping from basin $a$ to the outside of basin $a$ is*

$$
\tau = \frac{2\pi \sqrt{-\det(H_b)}}{(1 - \frac{d}{2\kappa})\sqrt{\det(H_a)}} \frac{1}{|H_{be}|} \left(1 + \frac{1}{\eta\kappa\sigma_e}\Delta L\right)^{\kappa - \frac{1}{2}}, \tag{25}
$$

*where $e$ indicates the most possible escape direction, $H_{be}$ is the only negative eigenvalue of $H_b$, $\sigma_e$ is the eigenvalue of $\Sigma_{g_a}$ corresponding to the escape direction and $\Delta L = L(b) - L(a)$.*

*Proof:* According to (Van Kampen, 1992), the mean escaping time $\tau$ is expressed as $\tau = \frac{P(w \in V_a)}{\int_\Omega J d\Omega}$, where $V_a$ is the volume of basin $a$, $J$ is the probability current that satisfies $-\nabla \cdot J(w, t) = \frac{\partial p(w, t)}{\partial t}$.

Under the low temperature assumption, the probability current $J$ concentrates along the direction corresponding the negative eigenvalue of $H_{be}$, and the probability flux of other directions can be ignored. Then we have

$$
\int_\Omega J d\Omega = J_e \cdot \int_\Omega \left(1 + \frac{1}{\eta\kappa}(w-b)^T (H_b \Sigma_g^{-1})^{\perp e}(w-b)\right)^{-\kappa + \frac{1}{2}} d\Omega, \tag{26}
$$

where $J_e = p(a) \cdot \frac{\eta(1 + \mu\sigma_e \Delta L(b))^{-\kappa + \frac{1}{2}} \sqrt{\mu\sigma_e|H_{be}|}}{2\sqrt{2}B(\frac{1}{2}, \kappa)}$ which is obtained by the calculation of $J_e$ for 1-dimensional case in the proof of Lemma 13, and $(\cdot)^{\perp e}$ denotes the directions perpendicular to the escape direction e.

Suppose $H_b \Sigma_g^{-1}$ are symmetric matrix. Then there exist orthogonal matrix $Q$ and diagonal matrix $\Lambda = diag(\lambda_1, \cdots, \lambda_d)$ that satisfy $H_b \Sigma_g^{-1} = Q^T \Lambda Q$. We also denote $v = Q(w-b)$.

We define a sequence as $T_k = 1 + \frac{1}{\eta\kappa} \cdot \sum_{j=k}^{d} \lambda_j v_j^2$ for $k = 1, \cdots, d$. As for the term $\int_\Omega \left(1 + \frac{1}{\eta\kappa}(w-b)^T (H_b \Sigma_g^{-1})^{\perp e}(w-b)\right)^{-\kappa+\frac{1}{2}} d\Omega$, we have

$$\int_\Omega \left(1 + \frac{1}{\eta\kappa}(w-b)^T (H_b \Sigma_g^{-1})^{\perp e}(w-b)\right)^{-\kappa+\frac{1}{2}} d\Omega$$

$$= \int (1 + \frac{1}{\eta\kappa} \cdot v^T \Lambda v)^{-\kappa+\frac{1}{2}} dw$$

$$= \int (1 + \frac{1}{\eta\kappa} \cdot \sum_{j\neq e}^{d} \lambda_j v_j^2)^{-\kappa+\frac{1}{2}} dv$$

$$= ((\eta\kappa)^{-1}\lambda_1)^{-\frac{1}{2}} \int T_2^{-\kappa+\frac{1}{2}} B(\frac{1}{2},\kappa) dv$$

$$= \prod_{j=0}^{d-2} ((\eta\kappa)^{-1}\lambda_j)^{-\frac{1}{2}} B(\frac{1}{2}, \kappa - \frac{j}{2})$$

$$= \prod_{j=0}^{d-2} ((\eta\kappa)^{-1}\lambda_j)^{-\frac{1}{2}} \cdot \frac{\sqrt{\pi^d}\Gamma(\kappa - \frac{d}{2})}{\Gamma(\kappa)}$$

$$= \frac{\sqrt{(\eta\kappa\pi)^{d-1}} \cdot \Gamma(\kappa - \frac{d-2}{2})}{\Gamma(\kappa + \frac{1}{2})\sqrt{\det((H_b \Sigma_g^{-1})^{\perp e})}}.$$

As for the term $P(w \in V_a)$, we have

$$P(w \in V_a) = \int_{V_a} p(w) dV = p(a) \int_{w \in V_a} \left(1 + (w-w^*)^T H_a \Sigma_g^{-1}(w-w^*)\right) dw \qquad (27)$$

$$= p(a) \cdot \frac{\sqrt{(\eta\kappa\pi)^d} \cdot \Gamma(\kappa - \frac{d}{2})}{\Gamma(\kappa)\sqrt{\det((H_a \Sigma_g^{-1}))}} \qquad (28)$$

where we use Taylor expansion of $L(w)$ near local minimum $a$.

Combined the results for $P(w \in V_a)$ and $J$, we can get the result. $\square$

### 7.3.2 FURTHER EXPLANATION ABOUT ASSUMPTION 1-3

We adopt the commonly used assumptions to analyze mean escaping time for dynamic system (Xie et al., 2020; Smith & Le, 2017; Zhou & Du, 2014). Assumption 2 can be replaced by weaker assumption that the system is quasi-equilibrium which is adopted in (Xie et al., 2020). For the differences between quasi-equilibrium and equilibrium, readers can refer to (Xie et al., 2020) for detailed discussions. Assumption 3 is commonly used (Xie et al., 2020; Zhou & Du, 2014). Under Assumption 3, the probability densities will concentrate around minima and the most possible paths. Assumption 3 will make the second order Taylor approximation more reasonable.

### 7.3.3 EXTENSION TO MORE COMPLEX DYNAMIC ON THE ESCAPING PATH

In Lemma 6, we assume that $C(w) = \sigma_{g_a} + \frac{2\sigma_{H_a}}{H_a}(L(w) - L(a))$ on the whole escaping path from $a$ to $b$ for ease of comparison and presentation. This assumption is not necessary and we can assume a different dynamic near saddle point $b$. Specially, we can assume the point $z$ is the midpoint on the most possible path beween $a$ and $b$, where $L(z) = (1-z)L(a) + zL(b)$. The dynamic with $C(w) = \sigma_{g_a} + \frac{2\sigma_{H_a}}{H_a}(L(w) - L(a))$ dominates the path $a \to z$ and the dynamic with $C(w) = \sigma_{g_b} + \frac{2\sigma_{H_b}}{H_b}(L(b) - L(w))$ dominates the path $z \to b$. Then only two things will be changed in proof of Lemma 6. First, we need to change the stationary distribution near saddle points according to its own dynamic in Eq.20. Second, we need to change the integral about probability density on

the whole path to sum of integrals on these two sub-paths. Similar proof techniques are adopted for analyzing escaping time of Langevin dynamic in proof of Theorem 4.1 in the work Xie *et al.* (2020). Since the proof is analogous, we omit the details here.

### 7.4 PAC-BAYES GENERALIZATION BOUND

We briefly introduce the basic settings for PAC-Bayes generalization error. The expected risk is defined as $\mathbb{E}_{x\sim\mathcal{P}(x)}\ell(w,x)$. Suppose the parameter follows a distribution with density $p(w)$, the expected risk in terms of $p(w)$ is defined as $\mathbb{E}_{w\sim p(w),x\sim\mathcal{P}(x)}\ell(w,x)$. The empirical risk in terms of $p(w)$ is defined as $\mathbb{E}_{w\sim p(w)}L(w) = \mathbb{E}_{w\sim p(w)}\frac{1}{n}\sum_{i=1}^{n}\ell(w,x_i)$. Suppose the prior distribution over the parameter space is $p'(w)$ and $p(w)$ is the distribution on the parameter space expressing the learned hypothesis function. For power-law dynamic, $p(w)$ is its stationary distribution and we choose $p'(w)$ to be Gaussian distribution with center $w^*$ and covariance matrix $I$. Then we can get the following theorem.

**Theorem 16** *(Theorem 8 in main paper) For $w \in \mathbb{R}^d$, we select the prior distribution $p'(w)$ to be standard Gaussian distribution. For $\delta > 0$, with probability at least $1 - \delta$, the stationary distribution of power-law dynamic has the following generalization error bound,*

$$\mathbb{E}_{w\sim p(w),x\sim\mathcal{P}(x)}\ell(w,x) \leq \mathbb{E}_{w\sim p(w)}L(w) + \sqrt{\frac{KL(p||p') + \log\frac{1}{\delta} + \log n + 2}{n-1}}, \qquad (29)$$

*where $KL(p||p') \leq \frac{1}{2}\log\frac{\det(H)}{\det(\Sigma_g)} + \frac{Tr(\eta\Sigma_g H^{-1})-2d}{4\left(1-\frac{1}{\kappa}\left(\frac{d}{2}-1\right)\right)} + \frac{d}{2}\log\frac{2}{\eta}$ and $\mathcal{P}(x)$ is the underlying distribution of data $x$.*

*Proof:* Eq.(29) directly follows the results in (McAllester, 1999). Here we calculate the Kullback–Leibler (KL) divergence between prior distribution and the stationary distribution of power-law dynamic. The prior distribution is selected to be standard Gaussion distribution with distribution density $p'(w) = \frac{1}{\sqrt{(2\pi)^d \det(I)}}\exp\{-\frac{1}{2}(w-w^*)^T I(w-w^*)\}$. The posterior distribution density is the stationary distribution for power-law dynamic, i.e., $p(w) = \frac{1}{Z}\cdot(1+\frac{1}{\eta\kappa}\cdot(w-w^*)^T H\Sigma_g^{-1}(w-w^*))^{-\kappa}$.

Suppose $H\Sigma_g^{-1}$ are symmetric matrix. Then there exist orthogonal matrix $Q$ and diagonal matrix $\Lambda = diag(\lambda_1, \cdots, \lambda_d)$ that satisfy $H\Sigma_g^{-1} = Q^T\Lambda Q$. We also denote $v = Q(w-w^*)$.

We have

$$\log\left(\frac{p(w)}{p'(w)}\right)$$

$$= -\kappa\log(1+\frac{1}{\eta\kappa}\cdot(w-w^*)^T H\Sigma_g^{-1}(w-w^*)) - \log Z + \frac{1}{2}(w-w^*)^T I(w-w^*) + \frac{d}{2}\log 2\pi$$

The KL-divergence is defined as $KL(p(w)||p'(w)) = \int_w p(w)\log\left(\frac{p(w)}{p'(w)}\right)dw$. Putting $v = Q(w - w^*)$ in the integral, we have

$$KL(p(w)||p'(w))$$

$$= \frac{d}{2}\log 2\pi - \log Z + \frac{1}{2Z}\int_v v^T v\left(1+\frac{1}{\eta\kappa}\cdot v^T\Lambda v\right)^{-\kappa}dv - \frac{1}{Z\eta}\int_v v^T\Lambda v\cdot(1+\frac{1}{\eta\kappa}\cdot v^T\Lambda v)^{-\kappa}dv, \tag{30}$$

where we use the approximation that $\log(1 + x) \approx x$. We define a sequence as $T_k = 1 + \frac{1}{\eta\kappa} \cdot \sum_{j=k}^{d} \lambda_j v_j^2$ for $k = 1, \cdots, d$. We first calculate the normalization constant $Z$.

$$
Z = \int (1 + \frac{1}{\eta\kappa} \cdot v^T \Lambda v)^{-\kappa} dw = \int (1 + \frac{1}{\eta\kappa} \cdot \sum_{j=1}^{d} \lambda_j v_j^2)^{-\kappa} dv
$$

$$
= ((\eta\kappa)^{-1}\lambda_1)^{-\frac{1}{2}} \int T_2^{-\kappa+\frac{1}{2}} B(\frac{1}{2}, \kappa - \frac{1}{2}) dv = \prod_{j=1}^{d} ((\eta\kappa)^{-1}\lambda_j)^{-\frac{1}{2}} B(\frac{1}{2}, \kappa - \frac{j}{2})
$$

$$
= \prod_{j=1}^{d} ((\eta\kappa)^{-1}\lambda_j)^{-\frac{1}{2}} \cdot \frac{\sqrt{\pi^d}\Gamma(\kappa - \frac{d}{2})}{\Gamma(\kappa)}
$$

We define $Z_j = ((\eta\kappa)^{-1}\lambda_j)^{-\frac{1}{2}} B\left(\frac{1}{2}, \kappa - \frac{j}{2}\right)$. For the third term in Eq.(30), we have

$$
2Z \cdot III
$$
$$
= \int_v v^T v (1 + \frac{1}{\eta\kappa} v^T \Lambda v)^{-\kappa} dv
$$
$$
= \int_{v_2,\cdots v_d} \int_{v_1} v_1^2 \left(1 + \frac{1}{\eta\kappa} \cdot v^T \Lambda v\right)^{-\kappa} dv_1 + Z_1 \left(\sum_{j=2}^{d} v_j^2\right)\left(1 + \frac{1}{\eta\kappa} \cdot \sum_{j=2}^{d} \lambda_j v_j^2\right)^{-\kappa+\frac{1}{2}} d_{v_2\cdots,v_d}
$$
$$
= \int_{v_2,\cdots v_d} T_2^{-\kappa} \int_{v_1} v_1^2 \left(1 + \frac{(\eta\kappa)^{-1}\lambda_1 v_1^2}{T_2}\right)^{-\kappa} dv_1 + Z_1 \left(\sum_{j=2}^{d} v_j^2\right)\left(1 + \frac{1}{\eta\kappa} \cdot \sum_{j=2}^{d} \lambda_j v_j^2\right)^{-\kappa+\frac{1}{2}} d_{v_2\cdots,v_d}
$$
$$
= \int_{v_2,\cdots,v_d} T_2^{-\kappa} \int \left(\frac{T_2}{(\eta\kappa)^{-1}\lambda_1}\right)^{\frac{3}{2}} y^{\frac{1}{2}} (1+y)^{-\kappa} dy + Z_1 \left(\sum_{j=2}^{d} v_j^2\right)\left(1 + \frac{1}{\eta\kappa} \cdot \sum_{j=2}^{d} \lambda_j v_j^2\right)^{-\kappa+\frac{1}{2}} d_{v_2\cdots,v_d}
$$
$$
= \int_{v_2,\cdots,v_d} ((\eta\kappa)^{-1}\lambda_1)^{-\frac{3}{2}} T_2^{-\kappa+\frac{3}{2}} B\left(\frac{3}{2}, \kappa - \frac{3}{2}\right) + Z_1 \left(\sum_{j=2}^{d} v_j^2\right)\left(1 + \frac{1}{\eta\kappa} \cdot \sum_{j=2}^{d} \lambda_j v_j^2\right)^{-\kappa+\frac{1}{2}} d_{v_2\cdots,v_d}
$$
$$
= (\frac{\lambda_1}{\eta\kappa})^{-\frac{3}{2}} B\left(\frac{3}{2}, \kappa - \frac{3}{2}\right) \int_{v_2,\cdots,v_d} T_2^{-\kappa+\frac{3}{2}} d_{v_2\cdots,v_d} + \int_{v_2,\cdots,v_d} Z_1 \left(\sum_{j=2}^{d} v_j^2\right)\left(1 + \frac{1}{\eta\kappa} \cdot \sum_{j=2}^{d} \lambda_j v_j^2\right)^{-\kappa+\frac{1}{2}} d_{v_2\cdots,v_d}
$$

For term $\int_{v_2,\cdots,v_d} T_2^{-\frac{1}{\kappa}+\frac{3}{2}} d_{v_2\cdots,v_d}$ in above equation, we have

$$
\int_{v_2,\cdots,v_d} T_2^{-\kappa+\frac{3}{2}} d_{v_2\cdots,v_d}
$$
$$
= \int_{v_3,\cdots,v_d} T_3^{-\kappa+2} ((\eta\kappa)^{-1}\lambda_2)^{-\frac{1}{2}} B\left(\frac{1}{2}, \kappa - 2\right) d_{v_3,\cdots,v_d}
$$
$$
= \int_{v_4,\cdots,v_d} T_4^{-\kappa+\frac{5}{2}} ((\eta\kappa)^{-1}\lambda_2)^{-\frac{1}{2}} ((\eta\kappa)^{-1}\lambda_3)^{-\frac{1}{2}} B\left(\frac{1}{2}, \kappa - \frac{5}{2}\right) B\left(\frac{1}{2}, \kappa - 2\right) d_{v_4,\cdots,v_d}
$$
$$
= \int_{v_d} T_d^{-\kappa+\frac{1}{2}+\frac{1}{2}\times d} \prod_{j=2}^{d-1} ((\eta\kappa)^{-1}\lambda_j)^{-\frac{1}{2}} \prod_{j=2}^{d-1} B\left(\frac{1}{2}, \kappa - (\frac{j}{2}+1)\right) d_{v_d}
$$
$$
= \prod_{j=2}^{d} ((\eta\kappa)^{-1}\lambda_j)^{-\frac{1}{2}} \prod_{j=2}^{d} B\left(\frac{1}{2}, \kappa - (\frac{j}{2}+1)\right)
$$

Let $A_j = ((\eta\kappa)^{-1}\lambda_j)^{-\frac{3}{2}} B\left(\frac{3}{2}, \kappa - (\frac{j}{2} + 1)\right)$. According to the above two equations, we can get the recursion

$$2Z \int v^T v T_1^{-\kappa} dv$$

$$= A_1 \cdot \int T_2^{-\kappa+\frac{3}{2}} + Z_1 \int_{v_2,\cdots,v_d} \left(\sum_{j=2}^{d} v_j^2\right) T_2^{-\kappa+\frac{1}{2}} d_{v_2\cdots,v_d}$$

$$= A_1 \cdot \int T_2^{-\kappa+\frac{3-1}{2}} d_{v_2\cdots v_d} + Z_1 \cdot A_2 \int T_3^{-\kappa+\frac{4}{2}} d_{v_3\cdots,v_d} + Z_1 Z_2 \int \left(\sum_{j=3}^{d} v_j^2\right) T_3^{-\kappa+\frac{1}{2}} d_{v_3\cdots,v_d}$$

$$= \sum_{j=1}^{d-1} A_j \prod_{k=1}^{j-1} Z_k \int T_{j+1}^{-\kappa+\frac{j+1+1}{2}} d_{v_{j+1},\cdots,v_d} + \prod_{k=1}^{d-1} Z_k \int v_d^2 T_d^{-\kappa+\frac{d-1}{2}} dv_d$$

$$= \sum_{j=1}^{d-1}(\frac{\lambda_j}{\eta\kappa})^{-\frac{3}{2}} B\left(\frac{3}{2}, \kappa - (\frac{j}{2}+1)\right) \prod_{k=1}^{j-1}(\frac{\lambda_k}{\eta\kappa})^{-\frac{1}{2}} B\left(\frac{1}{2}, \kappa - \frac{k}{2}\right) \prod_{s=j+1}^{d}((\frac{\lambda_s}{\eta\kappa})^{-\frac{1}{2}} \prod_{s=j+1}^{d} B\left(\frac{1}{2}, \kappa - (\frac{s}{2}+1)\right)$$

$$+ \prod_{j=1}^{d-1}(\frac{\lambda_j}{\eta\kappa})^{-\frac{1}{2}} B(\frac{1}{2}, \kappa - \frac{j}{2} - 1) \cdot (\frac{\lambda_d}{\eta\kappa})^{-\frac{3}{2}} B(\frac{3}{2}, \kappa - (\frac{d}{2}+1))$$

$$= \frac{\sqrt{\pi^d}\Gamma(\kappa - \frac{d}{2} - 1)Tr(H^{-1}\Sigma_g)}{2\Gamma(\kappa)\sqrt{(\eta\kappa)^{-(d+2)}\det(H^{-1}\Sigma_g)}}$$

We have

$$III = \frac{\sqrt{\pi^d}\Gamma(\kappa - \frac{d}{2} - 1)Tr(H^{-1}\Sigma_g)}{4\Gamma(\kappa)\sqrt{(\eta\kappa)^{-(d+2)}\det(H^{-1}\Sigma_g)}} \cdot \prod_{j=1}^{d}((\eta\kappa)^{-1}\lambda_j)^{\frac{1}{2}} \cdot \frac{\Gamma(\kappa)}{\sqrt{\pi^d}\Gamma(\kappa - \frac{d}{2})}$$

$$= \frac{\eta\kappa Tr(H^{-1}\Sigma_g)}{4(\kappa - \frac{d}{2} - 1)}$$

Similarly, for the fourth term in Eq.(30), we have $IV = \frac{\kappa d}{2(\kappa - \frac{d}{2} - 1)}$. Combining all the results together, we can get $KL(p||p') = \frac{1}{2}\log\frac{\det(H)}{(\eta\kappa)^d\det(\Sigma_g)} + \log\frac{\Gamma(\kappa)}{\Gamma(\kappa - \frac{d}{2})} + \frac{Tr(\eta\Sigma_g H^{-1}) - 2d}{4(1 - \frac{1}{\kappa}(\frac{d}{2} - 1))} + \frac{d}{2}\log 2$. Using the fact that $\log\frac{\Gamma(\kappa)}{\Gamma(\kappa - \frac{d}{2})} \le \frac{d}{2}\log\kappa$, we have $KL(p||p') \le \frac{1}{2}\log\frac{\det(H)}{\det(\Sigma_g)} + \frac{Tr(\eta\Sigma_g H^{-1}) - 2d}{4(1 - \frac{1}{\kappa}(\frac{d}{2} - 1))} + \frac{d}{2}\log\frac{2}{\eta}$.

## 7.5 IMPLEMENTATION DETAILS OF THE EXPERIMENTS

### 7.5.1 OBSERVATIONS ON THE COVARIANCE MATRIX

In this section, we introduce the settings on experiments of the quadratic approximation of covariance of the stochastic gradient on plain convolutional neural network (CNN) and ResNet. For each model, we use gradient descent with small constant learning rate to train the network till it converges. The converged point can be regarded as a local minimum, denoted as $w^*$.

As for the detailed settings of the CNN model, the structure for plain CNN model is $input \rightarrow Conv1 \rightarrow maxpool \rightarrow Conv2 \rightarrow maxpool \rightarrow fc1 \rightarrow Relu \rightarrow fc2 \rightarrow output$. Both $Conv1$ and $Conv2$ use $5 \times 5$ kernels with 10 channels and no padding. Dimensions of full connected layer $fc1$ and $fc2$ are $1600 \times 50$ and $50 \times 10$ respectively. We randomly sample 1000 images from FashionMNIST (Xiao *et al.*, 2017) dataset as training set. The initialization method is the Kaiming initialization (He *et al.*, 2015) in PyTorch. The learning rate of gradient descent is set to be $0.1$. After 3000 iterations, GD converges with almost $100\%$ training accuracy and the training loss being $1e^{-3}$.

As for ResNet, we use the ResNet-18 model (He *et al.*, 2016b) and randomly sample 1000 images from Kaggle's dogs-vs-cats dataset as training set. The initialization method is the Kaiming initialization (He *et al.*, 2015) in PyTorch. The learning rate of gradient descent is set to be $0.001$. After 10000 iterations, GD converges with $100\%$ training accuracy and the training loss being $1e^{-3}$.

We then calculate the covariance matrix of the stochastic gradient at some points belonging to the local region around $w^*$. The points are selected according to the formula: $w^*_{layerL} \pm (i \times Scale)$, where $w^*_{layerL}$ denotes the parameters at layer $L$, and $i \times Scale, i \in [N]$ determines the distance away from $w^*_{layerL}$. When we select points according to this formula by changing the parameters at layer $L$, we fixed the parameters at other layers. For both CNN model and ResNet18 model, we select 20 points by setting $i = 1, \cdots, 10$. For example, for CNN model, we choose the 20 points by changing the parameters at the $Conv1$ layer with $Scale = 0.001$ and $Conv2$ layer with $Scale = 0.0001$, respectively. For ResNet18, we choose the 20 points by changing the parameters for a convolutional layer at the first residual block with $Scale = 0.0001$ and second residual block with $Scale = 0.0001$, respectively.

The results are shown in Figure.1. The x-axis denotes the distance of the point away from the local minimum and the y-axis shows the value of the trace of covariance matrix at each point. The results show that the covariance of noise in SGD is indeed not constant and it can be well approximated by quadratic function of state (the blue line in the figures), which is consistent with our theoretical results in Section 3.1.

### 7.5.2 Supplementary Experiments on Parameter Distributions of Deep Neural Networks

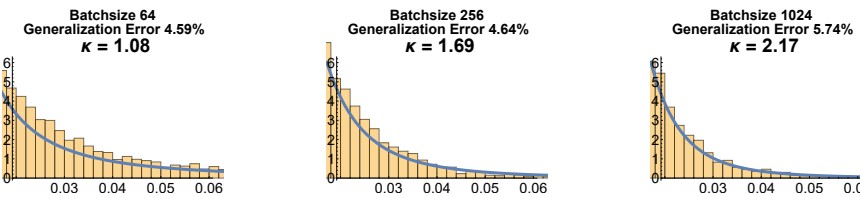

Figure 6: A close-up of right tail distribution of the result for ResNet18 in Figure. 3(a), which could help to observe the heavy-tailed properties among different batchsize.

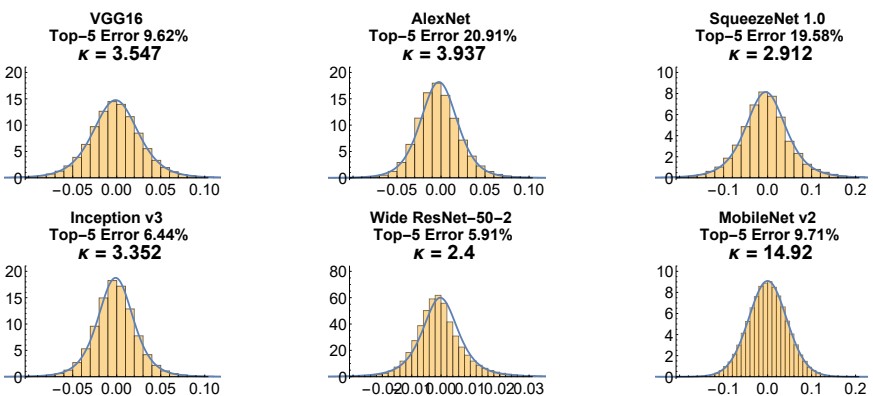

Figure 7: Approximating distribution of parameters (trained by SGD) by power-law dynamic. These networks use pre-trained models offered by PyTorch, and all of them are pre-trained on ImageNet dataset using SGD. The second line in each title shows Top-5 test error and the third line shows approximated tail-index $\kappa$.

For Figure. 3(a), we train LeNet-5 on MNIST dataset using SGD with constant learning rate $\eta = 0.03$ for each batchsize till it converges. Parameters are $conv2.weight$ in LeNet-5. For Figure. 3(b), we train ResNet-18 on CIFAR10 using SGD with momentum. We do a $RandomCrop$ on training set scaling to $32 \times 32$ with $padding = 4$ and then a $RandomHorizontalFlip$. In training, momentum is set to be $0.9$ and weight decay is set to be $5e - 4$. Initial learning rate in SGD is set to be $0.1$ and we using a learning rate decay of $0.1$ on $\{150, 250\}$-th epoch respectively. We train it until converges after 250 epoch. Parameters are $layer1.1.conv2.weight$ in ResNet-18.

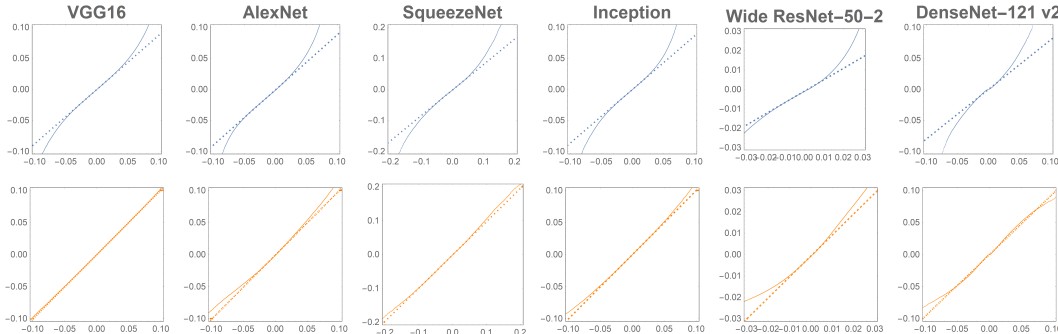

Figure 8: Comparison between Q-Q plots of network parameters versus normal distribution and power-law distribution. **(upper):** Q-Q plots of parameters versus normal distribution. **(bottom):** Q-Q plots of parameters versus power-law distribution.

We also observe the parameter distribution on many pretrained models. Details for pre-trained models can be found on `https://pytorch.org/docs/stable/torchvision/models.html`. Figure.7 shows the distribution of parameters trained by SGD can be well fitted by power-law distribution. Parameters in this figure are all randomly selected to be *features.10.weight*, *features.14.weight*, $features.5.expand3 \times 3.weight$, $Mixed\_6d.branch7 \times 7\_3.conv.weight$, $layer4.2.conv3.weight$ and $features.denseblock2.denselayer1.conv2.weight$ for VGG-16, AlexNet, SqueezeNet 1.0, Inception v3, Wide ResNet-50-2 and DenseNet-121 respectively.

A Q-Q plot is created by plotting quantiles of two probability distributions against one another, which can provide an assessment of "goodness of fit" by how much the solid line close to the dashed line. From Figure.8, it is clear that the solid lines in bottom pictures are closer to dashed lines on most cases, which indicates network parameters can be better fitted by power-law distribution. Moreover, solid lines in the upper plots severely deviate from dashed lines on the tail of distribution but those in the bottom plot do not, which means the distribution of parameters is indeed heavy-tailed.

### 7.5.3 FURTHER EXPLANATION ON EXPERIMENTS IN SECTION 5.2

As for the experiments for 2-D model, we also calculate coefficient of the second-order term for the quadratic curve shown in Figure.4(b), and its value is roughly 30, which matches the result in Figure.4(c) in the sense that the result for SGD is similar with the result for power-law dynamic with $\lambda_1 \approx 32$.

### 7.5.4 ESCAPING EFFICIENCY ON NEURAL NETWORK

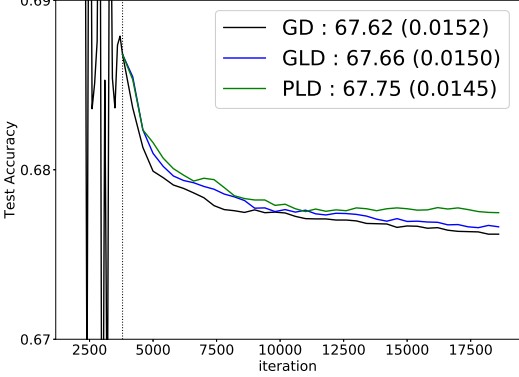

Figure 9: Escaping experiment on corrupted FashionMNIST. Test accuracy versus iteration after pretraining by GD. Model is pretrained by GD before the vertical dashed line and continued by GD, GLD and PLD (ours). Numbers in brackets are expected sharpness after model converging.

We follow the settings in (Zhu *et al.*, 2019). For convenience of the readers, here we give the details of this setting again. We use corrupted FashionMNIST dataset which contains 1000 images with correct labels and another 200 images with random labels to be training data. A small LeNet-like network with 11,330 parameters is used. Firstly we run the full gradient decent to reach the parameters $w_*$ near the global minima. Then we continue training using both Langevin dynamic(GLD) and power-law dynamic(PLD). Following Zhu's setting, the learning rates for GD, GLD and PLD are $\eta_{GD} = 0.1, \eta_{GLD} = 0.07$ and $\eta_{PLD} = 0.07$, respectively. For GLD, noise std $\sigma = 10^{-4}$ as Zhu already tuned. For our PLD, $w_{t+1} = w_t - \eta\nabla L(w_t) + \eta \cdot \alpha\nabla L(w_t) \odot \sqrt{1 + \beta(w_t - w^*)^2} \odot \xi$, where $\alpha, \beta$ are hyperparameters, $\xi \sim \mathcal{N}(0, I)$, and $\odot$ stands for Hadamard product. Here we select $\alpha = 2.4, \beta = 2$ after grid search. Expected sharpness is measured as $\mathbb{E}_{\nu \sim N(0, \delta^2 I)}[L(w + \nu)] - L(w)$ where $\delta = 0.01$, and the expectation is computed by average on 1000 times sampling.

The numbers at the first column of the legend show the test accuracy and the numbers in the bracket show the sharpness of the model trained by the three algorithms. From Figure 9, we can conclude that PLD generalizes better than GLD and GD. Moreover, PLD can find flatter critical points than GLD and GD.

### 7.5.5 COMPARISON OF MEAN ESCAPING TIME WITH DIFFERENT BARRIER HEIGHTS

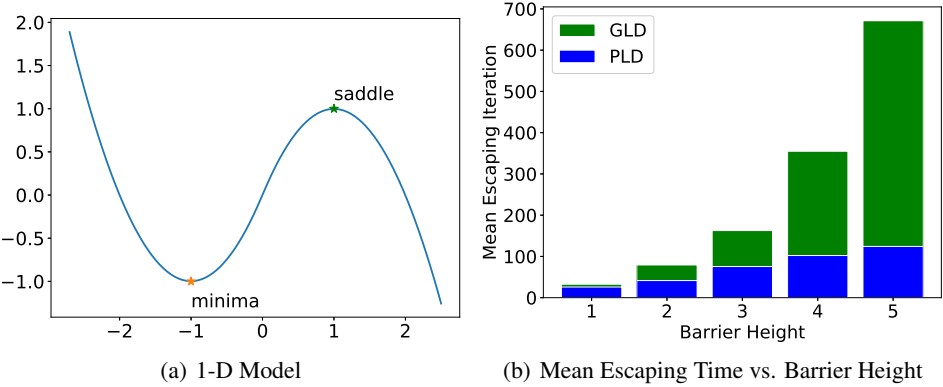

(a) 1-D Model          (b) Mean Escaping Time vs. Barrier Height

Figure 10: **(a)**: Loss curve $L(w)$ for 1-D model. **(b)**: Mean escaping time versus different barrier heights. Mean escaping time is computed by average on 100 rounds, in which we record the number of iterations when firstly escaping from the saddle point.

We design this 1-dimensional model to help to validate the theoretical results of escaping time in Table 1. Loss function $L(w) = \frac{1}{n}\sum_{i=1}^{n}\ell(w - x_i)$, where $\ell(w) = \begin{cases} w^2 + bw & , w < 0 \\ -w^2 + bw & , w \geq 0 \end{cases}$ and $x_i \sim \mathcal{N}(0, 0.05)$. $L(w)$ is plotted in Figure.10(a), and we can adjust barrier height through parameter $b$ in $\ell(w)$ without changing the Hessian on minima $w^*$ and saddle point.

For power-law dynamic (PLD), $w_{t+1} = w_t - \eta\nabla L(w_t) + \eta\lambda_2\sqrt{1 + \lambda_1(w_t - w^*)^2} \odot \xi$, where $\lambda_1, \lambda_2$ are hyperparameters, $\xi \sim \mathcal{N}(0, I)$, and $\odot$ stands for Hadamard product. Here we let $\lambda_1 = 1, \lambda_2 = 4$. For Langevin dynamic (GLD), we set noise std $\sigma = 4$ in consistence with PLD. Learning rate $\eta = 0.1$ for both methods. We initialize $w_0 = w^*$ and apply both methods on $L(w)$ with different barrier heights. Then we record the number of iterations $t$ when $w_t$ firstly escaping from the barrier. We repeat this procedure 100 rounds for each method and each barrier height and utilize the average to estimate the mean escaping time, of which the results are shown in Figure.10(b).

From Figure.10(b), the mean escaping time of GLD grows much faster than PLD along barrier height, which validates that power-law dynamic improves the order of barrier height compared with Langevin dynamic.

