# OpenReview forum: "Dynamic of Stochastic Gradient Descent with State-dependent Noise"
_ICLR.cc/2021/Conference — Reject_

### Official Review · AnonReviewer1 · 2020-10-13
**Novel perspective, but left with much ambiguous analysis and insufficient empirical justifications**

**Rating:** 5
**Confidence:** 5

**Review:**

This paper proposes to use power-law dynamics to approximate the state-dependent gradient noise in SGD, and analyses its escaping efficiency compared with previous dynamics.

Strength:
1.	To the best of my knowledge, it is novel to use power-law dynamics to analyze the state-dependent noise in SGD.
2.	Still with strong assumptions on covariance structure, the analytical results based on power-dynamics are interesting. For example, it indicates that so-called kappa distribution highly depends on the fluctuations to the curvature over the training data. This is consistent with following work. So I suggest authors provide some discussion with the following work.
Wu et.al 2018. How sgd selects the global minima in over-parameterized learning: A dynamical stability perspective. In Advances in Neural Information Processing Systems (pp. 8279-8288).

Weakness & Issues
1.	The analytical results seem that they strongly depend on the covariance structure assumption, i.e. C(w) is diagonally dominant according to empirical observation. Does it have any theoretical justifications, or even in simplified cases?
2.	The delivered PAC generalization bound and the followed analysis are a little ambiguous.  Firstly, in current deep learning theory community, the relationship between flatness (even how to define a proper flatness) and generalization is still mysterious and controversial, which depends many factors. This work uses one type of flatness measure, the determinant of H, and shows that flatter minima generalize better by only considering the KL term. However, the first term also includes the Hessian and might also affect generalization bound. Thus, the conclusion appears a little problematic.
The authors said that generalization error will decrease w.r.t. kappa’s increase and infinite kappa results in Langevin dynamics. Then the question is what are the difference between the power-law dynamics and Langevin dynamics in term of generalization?
My view on the ambiguous analysis is that the authors attempt to answer extremely challenging questions but left with many questionable concerns.
3.	The experiments might not be sufficient.
I don’t think fitting the parameter distribution according to limited empirical observations is an appropriate way to make justifications. At least, from visual observation, there are many other alternatives besides power-law distribution to fit, as Fig 3 shows.
About comparing the escaping efficiency, the result only shows the success rate, and the evidence about the polynomial and exponential difference should be provided. Also, practical networks and datasets should also be considered to provide more strong evidence.

If the authors can resolve these issues carefully, I would raise the score.

Typos
“Eq. 4” should be “Eq.3” below equation 3

---

> ### Author Response · Authors · 2020-11-20
> **Response to Reviewer #1**
>
> Thank you very much for your valuable comments and constructive suggestions on experiments. The following are our responses.
> 1.	$C(w)$ is diagonally dominant according to empirical observation. Does it have any theoretical justifications, or even in simplified cases?
>
> “Diagonally dominant” indicates that the variance of each dimension of stochastic gradient $\tilde{g}(w)$ is larger than the covariance of two different dimensions of stochastic $\tilde{g}(w)$. As you suggested, we have added a simplified example of deep neural network at the initialization to justify this assumption in Appendix 7.2. The example shows that, the independence of different dimensions of stochastic gradient comes from the random initialization of non-overlapped parameters. Please kindly check Appendix 7.2 for the details.
>
> 2.	However, the first term also includes the Hessian and might also affect generalization bound.
> What are the differences between the power-law dynamics and Langevin dynamics in terms of generalization?
>
> a.	We are not sure that what is the first term you refer to. We guess that you mean the expected training loss term $\mathbb{E}_{w\sim p(w)}L(w)$.
>
> In Theorem 8, the generalization error means the difference between the training and test loss, i.e.,  $\mathbb{E}_{w,x}\ell(w,x)-{\mathbb{E}_w}L(w)$. Therefore, only the term contained KL divergence at the right side of the inequality is the upper bound of generalization error.
>
> Indeed, the Hessian will also influence the expected training loss (i.e., $\mathbb{E}_{w\sim p(w)}L(w)$), however, many optimization algorithms have enough ability to achieve low training loss in the over-parameterized regime. An example in Figure 3 in [1] shows that both Langevin dynamic with constant diffusion coefficient and SGD can achieve low training loss, but the test accuracy of Langevin dynamic is lower than SGD. Therefore, researchers mainly care about the generalization error [2].
>
> b.	 Our generalization bound shows that the generalization error of power-law dynamic is smaller than that of Langevin dynamic. As discussed in paragraph after Theorem 8, when kappa goes to infinity, the result goes to that for Langevin dynamic, which is consistent to the results in [2].
>
> 3.	At least, from visual observation, there are many other alternatives besides power-law distribution to fit, as Fig 3 shows.
>
> We have added the Q-Q plot (quantile-quantile plot) for power-law distribution and Gaussian distribution in Figure 8 in Appendix. Figure 8 shows that power-law distribution fits distribution of parameters better than Gaussian distribution.
>
> 4.	About comparing the escaping efficiency, the result only shows the success rate, and the evidence about the polynomial and exponential difference should be provided. Also, practical networks and datasets should also be considered to provide more strong evidence.
>
> a.	As you suggested, we conduct experiments to compare the escaping time with respect to different heights of barrier for power-law dynamic. The results are added in Appendix 7.5.5. Results in Figure 10(b) can support the theoretical results about mean escaping time of power-law dynamic and Langevin dynamic with respect to different barrier height.
>
> b.	We add the experiments on the escaping efficiency on practical networks in Appendix 7.5.4. We first use GD to train the network till it converges. Then the stopped point can be regarded as a local minimum. Then we use simulations of Langevin dynamic and power-law dynamic to continue the training. Results show that power-law dynamic can stopped at a flatter minimum and achieve higher test accuracy than Langevin dynamic.
>
> 5.	Thank you for recommending the paper [Wu et.al 2018]. We cite this reference in related works in the updated version.
>
> [1] Zhu, Zhanxing, Wu, Jingfeng, Yu, Bing, Wu, Lei, & Ma, Jinwen. 2019.  The anisotropic noise instochastic gradient descent: Its behavior of escaping from sharp minima and regularization effects.Pages 7654–7663 of: Proceedings of International Conference on Machine Learning.
>
> [2] He, Fengxiang, Liu, Tongliang, & Tao, Dacheng. 2019a. Control Batch Size and Learning Rate to Generalize Well: Theoretical and Empirical Evidence.Pages 1141–1150 of: Advances in NeuralInformation Processing Systems.

---

### Official Review · AnonReviewer2 · 2020-10-28
**Interesting theoretical results, though presentation could be improved**

**Rating:** 6
**Confidence:** 4

**Review:**

Summary: the paper studies the effect of SGD noise near a local minimum of the loss by using a novel Taylor expansion to estimate the distribution of gradient noise in the neighborhood of that minimum. They use this to derive closed-form equations describing the distribution of the iterates, which they use to characterize properties such escaping times and generalization.

Pros:
- To my knowledge, the mathematical analysis appears to be quite novel and insightful. It is particularly interesting that the authors show that the second order effect of the SGD noise in the Hessian induces a power law distribution over the iterates.
- Some empirical support is provided for the theory.

Cons:
- In general, a clearer statement (and justification) of the assumptions is required. For example, it appears to be implicit throughout the paper that we only consider the neighborhood of a local minimum, so the analysis is essentially for a quadratic in this neighborhood. This should be stated more explicitly.
- I also have some concerns about mathematical precision in the theorem statements. It is sometimes unclear which computations are rigorous equalities and which are not - for example, in Lemma 6 about escaping times, exact equality is used. However, the proof relies on Taylor expansion and uses approximate equalities in the steps. This is potentially misleading.

In general, the results seem interesting, and it is understandable that certain assumptions/heuristics must be used because this area of research is technically challenging. However, I would like to see the clarity of the presentation improved before recommending acceptance.

I have more specific questions regarding the details in the paper below:
- Could the authors elaborate on the relationship between kappa and generalization? From the paper my understanding was that smaller kappa meant flatter curvature and better generalization, but this doesn't seem to be supported by Figure 3.
- How is the value of kappa contained in Figure 3? Is it computed via computing the Hessian and its covariance or chosen to best fit the histograms in the figure?
- Eqn 6: It seems like this closed form computation is specifically for the case when the function is quadratic (e.g. we take 2nd order Taylor approximation around a local min. Can the authors confirm?) If this is the case, what happens to the dependency on w - w^* and why is there no such explicit term in eqn 6? It would appear that g(w) should depend on (w - w^*).
- In the overparameterized regime, it would appear that \sigma_g could go to 0 if each training example is overfit by the model. It appears that plugging in \sigma_g = 0 would introduce some degeneracy in equation 6 and 7, however. Can the authors comment on this?
- Intuition on the term \sigma_H: what do we expect this to look like in practice and do the authors have a sense on whether this term only matters around local minimum?
- The definition of \Sigma_H in the multivariate case: in the first paragraph of section 3, the definition on the LHS has no mention of i, j but the RHS does.
- Assuming the signal to noise ratio of \tilde{H} can be characterized by a scalar - why is this assumption reasonable?

*********
EDIT: Changed my score from 5 to 6 after the author response/revision.

---

> ### Author Response · Authors · 2020-11-20
> **Response to Reviewer #2**
>
> Thank you very much for the constructive comments for us to improve the mathematical precision and the recognition for the novelty and the insights of our work. We carefully revised the paper according to your comments. The revisions mainly include the followings:
> a) To make a clearer statement of the assumptions, we reorganized all the assumptions for escaping time and move them to the beginning of section 4. Based on the assumptions, equality in Lemma 6 is established.  b) We gave the expression of $C(w)$ in the theorems to make them self-contained.  c) We unified the notation of $\Sigma_H$.  d) We provided more justifications about the assumption on signal to noise ratio of \tilde{H} in Appendix 7.2.
>
> Regarding the specific questions, we have the following responses.
> 1.	"Relationship between kappa and generalization. From the paper my understanding was that smaller kappa meant flatter curvature and better generalization, but this doesn't seem to be supported by Figure 3."
>
> In Figure3(b), when kappa=1.08 with small batch size 64,  training accuracy is lower because of high magnitude of noise. To remove the influence of the training and better illustrate the relationship between kappa and generalization, we report the gap between test error and the training error in the updated version.  All the results are consistent with our theory.
>
> 2.	How is the value of kappa contained in Figure 3?
>
> The values of kappa in Figure 3 is chosen to best fit the histograms in the figure.
>
> 3.	what happens to the dependency on $w - w^*$ and Why is there no such explicit term in eqn6?
>
> As shown by the notations defined in the end of Sec 3.1, $C(w)$ depends on $w-w^*$. To make Theorem 2 self-contained, we have added the expression of $C(w)$ after eqn6 in the updated version.
>
> 4.	In the overparameterized regime, it would appear that $\sigma_g$ could go to 0 if each training example is overfit by the model.
>
> Thanks for raising this interesting question. In overparameterized regime, $\sigma_g$ could go to 0 when the value of loss goes to 0. At the same time, $\sigma_H$ goes to zero and $C(w)$ goes to zero. The dynamic will be static and cannot escape from the local minima if $C(w)$ equals to zero.  We added a footnote for this case in the updated version.
>
> 5.	Intuition on the term $\sigma_H$: what do we expect this to look like in practice and do the authors have a sense on whether this term only matters around local minimum?
>
> Intuitively, $\sigma_H$ measures the variance of the flatness. Both the averaged flatness (i.e., $H$) and the variance of the flatness (i.e., $\sigma_H$) will influence the dynamic behavior. In our current theoretical analyses, $\sigma_H$ mainly makes sense around local minima or saddle points.
>
> Although the theory is mainly focused on local minimum [1], in practice, it inspires us to inject state-dependent noise (instead of constant noise) into the algorithm on its whole optimization path. That is to say, $\sigma_H$ can be leveraged to help optimization algorithm escape critical points including local minima and saddle points efficiently. For example, when GD or large batch SGD stops, we can inject noise with variance $\sigma_g+\sigma_H(w-w^*)^2$ in them. We have done an experiment to demonstrate its efficiency. Please refer Appendix 7.5.4.
>
> 6.	Assumption on the signal to noise ratio of \tilde{H} – can be characterized by a scalar.
>
> This assumption is made for ease of theoretical analyses. Similar assumption is adopt to analyze high dimensional Hessian in [2] where they only consider the fluctuation of the largest eigenvalue of Hessian.
> In practice, we expect that the fluctuation of \tilde{H} follows: for small eigenvalues in $H$, their variances are also small. It results in that parameters move only in a low-dimensional space as discussion in section 5 in [3]. So, we use a scalar to characterize the ratio of $\Sigma_H$ and $H$ to roughly reflect the positive correlation between $H$ and $\Sigma_H$. Besides this assumption, we provide results based on much weaker assumption in proposition 12 in Appendix 7.2.
>
> Making this assumption also has practical benefit, i.e., we can use H and the scalar \kappa to   characterize the dynamic without calculating $\Sigma_H$. The analyses on escaping time and generalization for this dynamic show that this dynamic can generalize better than Langevin.
>
>
> [1]  Zhu, Zhanxing, Wu, Jingfeng, Yu, Bing, Wu, Lei, & Ma, Jinwen. 2019.  The anisotropic noise instochastic gradient descent: Its behavior of escaping from sharp minima and regularization effects. ICML2019
>
> [2] Wu et.al 2018. How sgd selects the global minima in over-parameterized learning: A dynamical stability perspective. Neurips2018
>
> [3] Xie, Zeke, Sato, Issei, & Sugiyama, Masashi. 2020. A Diffusion Theory for Deep Learning Dynamics:
> Stochastic Gradient Descent Escapes From Sharp Minima Exponentially Fast. arXiv preprint
> arXiv:2002.03495.

---

> > ### Comment · AnonReviewer2 · 2020-11-23
> > **Thanks for the clarification and revisions**
> >
> > Thank you for the clarifications and revisions. As the revision has improved the clarity of the results in the paper, I will increase my score slightly. I suggest that for future revisions, the authors continue to improve the clarity in their presentation of the results.

---

> > > ### Author Response · Authors · 2020-11-24
> > > **Thank you for raising the score**
> > >
> > > Thank you for raising the score. We’re glad to see that our response has addressed your concerns. We will continue to polish the paper as you suggested.

---

### Official Review · AnonReviewer3 · 2020-10-28

**Rating:** 6
**Confidence:** 3

**Review:**

### Summary
This paper proposes an analysis for an approximate dynamic of SGD which captures the heavy-tailed noise distributions seen practically at local minima. The authors derive this new dynamic (which they call Power-law dynamic) using basic principles and the assumption that the noise variance depends on the state. The dynamics becomes a modified Langevin equation. They prove also that the expected time to escape a barrier is polynomial in the parameters, as well as a generalization error bound.

### Strong/Weak points
- The paper is built up on simple principles
- It first gives a one-dimensional analysis then generalize, helping the reader to understand
- Except a few points, in general the paper is well-written.
- The assumptions made are somewhat strong and may not hold in some cases, see below.

In general I have a tendency to accept this paper. Even though there are crucial assumptions that are made, it can be considered as a first step towards a more rigorous and general argument.

Here are a few points that I have problems with in the paper:
- On page 3, paragraph 2, it is written that the solution of Langevin equation is Gaussian distribution. What does it mean? The solution of a SDE is a Markov process, and considering the distribution of the process at time $t$, it is not necessarily Gaussian; the Fokker-Planck equation governs the change of distribution, having the Gibbs distribution as its stationary distribution, which is not Gaussian in general.
- The whole argument is made through assuming that near the basin, everything is quadratic (not approximately, equal!). This is completely reflected in Proposition 1 and the further analysis.
- In Theorem 4, it is not stated that $H = H(w^*)$, and I don't see why should one be interested when $w\to\infty$? This is because we are talking about an $\epsilon$ ball around $w^*$ and tending $w \to \infty$ has no meaning.... Maybe I am missing something here? Also, it seems that the distribution is defined only for positive $w$.
- In the argument for Section 4 (escaping), it is assumed that the basin is quadratic and __stays__ quadratic, even when it reaches the saddle point. I find this assumption flawed, or I am missing something.
- At the bottom of page 3, it is said "in this case, .... is satisfied", while I think it should be "not satisfied".
- On page 4, the notion $\rightarrow_p$ is used for convergence in distribution, which is not usual and is reserved for convergence in probability.

---

> ### Author Response · Authors · 2020-11-20
> **Response to Reviewer #3**
>
> Thank you very much for your supportive review and constructive comments. Especially thanks for recognizing our work along this research direction. Our results are established by assuming that the loss is equal to quadratic function near the critical points. To make this explicit, we re-organize all the assumptions for escaping time and move them to the beginning of section 4. Thanks for pointing out the typos. We have revised them in the updated version. Here are our responses to your other questions about the paper. If any further confusing, please feel free to let us know.
> 1. “The solution of Langevin equation is Gaussian distribution.” What does it mean?
>
> Thanks for your careful reading. As defined in Eqn (1), the diffusion term in Langevin dynamic (investigated in machine learning) is constant. The solution of the (specific) Langevin equation with constant diffusion is Gaussian process. Its stationary distribution is Gaussian distribution. We revised this claim to be accurate in the updated version.
>
> 2. In Theorem 4, it is not stated that $H=H(w^*)$. This is because we are talking about an $\epsilon$-ball around $w^*$ and tending $w\rightarrow\infty$ has no meaning.... Also, it seems that the distribution is defined only for positive w.
>
> a. We have added “$H=H(w^*)$” in Theorem 4 in the updated version.
>
> b. In the updated version, instead of taking $w\rightarrow\infty$, we discuss about the decreasing rate of p(w) in the region $[w^*-\epsilon, w^*+\epsilon]$, which also indicates power-law distribution is less concentrated in the quadratic basin and heavy-tailed.
>
> c. The distribution in Eqn (9) is defined for $w\in[w^*-\epsilon, w^*+\epsilon]$, not only for positive w. It is symmetric and positive with respect to $(w-w^*)$.
>
> 3. “it is assumed that the basin is quadratic and stays quadratic, even when it reaches the saddle point.”
>
> We assume that the loss surface near critical points (including local points and saddle points) can be approximated by the second order Taylor expansion. For saddle points, the Hessian has one negative eigenvalue. Thus, the loss surface along the whole escaping path from a local point to the saddle point near it is not quadratic.

---

### Official Review · AnonReviewer4 · 2020-11-01
**Review of Dynamic of Stochastic Gradient Descent with State-dependent Noise**

**Rating:** 5
**Confidence:** 3

**Review:**

This paper proposes power-law dynamic of SGD which considers state-dependent noise. The power-law distributed derived from this  dynamic explains the heavy-tailed distribution of parameters trained by SGD. Besides, this dynamic also shows efficiency of escaping local minima.

Concerns:
1. The proof of theorem 2 is not provided in the appendix. But I doubt if C(w) is well-defined. It is not clear how w* is selected considering there are multiple local minima. It does not make sense to me if w* is fixed when taking x-->\infty, as the quadratic approximation should be used in the neighborhood of w*.

2. The escaping efficiency of the power-law dynamic is only analyzed in low-dimension case. I wonder how if performs in high-dimensional space. Does it provide more benefits than Langevin/alpha-stable dynamic in the expense of calculating sigma_g and sigma_H.

Minor comments:
1. I think  [Li et al., 2017] also proposed state-dependent noise in Theorem 1.

---

> ### Author Response · Authors · 2020-11-20
> **Response to Reviewer #4**
>
> Thank you very much for the valuable comments. Please check the proof of Theorem 2 in Appendix 7.1 in the updated version.
> Here are our responses to your two concerns.
> 1. “It is not clear how w* is selected considering there are multiple local minima. It does not make sense to me if w* is fixed when taking w-->\infty”
>
> Our analyses are established in the local region of the local minima and thus w^* can be any fixed local minima. In the updated version, instead of taking $w\rightarrow\infty$, we discuss about the decreasing rate of p(w) in the region $[w^*-\epsilon, w^*+\epsilon]$, which indicates power-law distribution is less concentrated in the quadratic basin and heavy-tailed.
>
> 2. “The escaping efficiency of the power-law dynamic is only analyzed in low-dimension case. Does it provide more benefits than Langevin/alpha-stable dynamic in the expense of calculating sigma_g and sigma_H."
>
> a) Please refer Theorem 7 in our submission for our results in multi-dimensional case. The result for multi-dimensional case also inherits the benefit of power-law dynamic in 1-dimensional case. The only difference is that dimension d appears in the denominator and kappa needs to be larger than d/2 in multi-dimensional case. As discussed in the remark after theorem 7, H is observed to be highly degenerate in deep learning [1]. Thus, d can be replaced by the number of dimensions whose corresponding eigenvalues are significantly larger than zero, which is much smaller.
>
> b) As Langevin dynamic with constant diffusion does not generalize well (shown in Figure 1 in [2]), in practice, our theory motivates us to inject state-dependent noise to GD or large batch SGD to improve its generalization. Although calculating the exact value of \sigma_H may be costly, we will investigate efficient approximation of it in future work.
>
> $\textbf{For your minor comment}$, although the work [Li et al., 2017] formulates the dynamic of SGD as SDE with state-dependent diffusion, all the theoretical analysis including “optimal control of learning rate and momentum” are derived under the assumption that C(w) is locally constant. We cite and discuss this paper in related work in the updated version.
>
> [1] Levent Sagun, Leon Bottou, Yann LeCun. 2017 Eigenvalues of the Hessian in Deep Learning: Singularity and Beyond
> [2] Zhu, Zhanxing, Wu, Jingfeng, Yu, Bing, Wu, Lei, & Ma, Jinwen. 2019.  The anisotropic noise instochastic gradient descent: Its behavior of escaping from sharp minima and regularization effects.Pages 7654–7663 of: Proceedings of International Conference on Machine Learning.

---

### Author Response · Authors · 2020-11-20
**Paper Revision**

We appreciate all the reviewers for their great efforts on our paper and the constructive comments to us.
We have updated our manuscript according to reviewers’ comments. The changes we made mainly include the followings.
1. Assumptions: we reorganize all the assumptions for escaping time and move them to the beginning of section 4.
2. Theorem 2: we give the expression of C(w) in the theorem to make it self-contained and put its proof in Appendix 7.1. We discuss the decreasing rate of p(w) in the local region of $w^*$ instead of taking $w\rightarrow\infty$, which indicates power-law distribution is less concentrated in the quadratic basin and heavy-tailed.
3. Multi-dimensional case: We provide more explanations about the results for multi-dimensional case in remark after Theorem 7 and the assumption on $\kappa$ in Appendix 7.2.
4. Experiments: we add the discussion for Figure 3(b) in Sec 5.1. We illustrate that power-law distribution can better fit the parameter distribution than Gaussian in Figure 8 in Appendix 7.5.2. We add the experiments on the escaping efficiency on practical networks in Appendix 7.5.4.
5. References and typos: we modify the citations for Langevin dynamic and cite [Wu, et al, 2018]. We correct the typos.

---

### Decision · Program_Chairs · 2021-01-07
**Final Decision**

**Decision:**

Reject

**Comment:**

The average review rating is 5.5 which means it’s somewhat borderline. One of the reviewers planned to increase the score but apparently didn’t do so formally. A subset of the main pros and cons the reviewers pointed out are:

Pros:
“Some empirical support is provided for the theory.”
“ It is particularly interesting that the authors show that the second order effect of the SGD noise in the Hessian induces a power law distribution over the iterates.”

Cons:
“The escaping efficiency of the power-law dynamic is only analyzed in low-dimension case. ...” The author responded that Theorem 7 proves the multi-dimensional case. But the AC noted that it’s very likely that escaping time is exponential in dimension (because kappa needs to be larger than d as the author noted and the det() might also be exponential in d. The author did say in the revision that the dimension should be considered as the effective dimension of the hessian, but the AC couldn’t find a formal argument about it.)
“The assumptions made are somewhat strong and may not hold in some cases...”

The reviewers also had a few clarity questions which the author addressed in revisions with re-organized writing. The AC weighed the pros and cons and found that the unclarity and potential exponential escaping time in the multi-dimensional case outweigh the pros.